# An Exterior Method for
# Nonnegative Matrix Factorization

Qiujing Lu [* 1]   Tonmoy Monsoor [* 1]   Ehsan Ebrahimzadeh [2]   Kartik Sharma [1]   Vwani Roychowdhury [1]

## Abstract

Nonnegative matrix factorization (NMF) seeks a low-rank approximation $X \approx UV^T$ with nonnegative factors and is commonly solved using *interior* methods that enforce feasibility throughout optimization. We show that such constraint-driven approaches can impede progress in the nonconvex landscape, leading to slow convergence or convergence to suboptimal stationary points. We propose an *exterior* framework for NMF (eNMF) that separates low-rank approximation from nonnegativity enforcement. Our method initializes from the optimal unconstrained factorization and introduces a rotation procedure that maps unconstrained factors to an exterior point closest to the nonnegative orthant. This viewpoint yields an algorithmic framework in which simple iterative updates converge to KKT-satisfying stationary points on the boundary of the positive orthant. The exterior formulation also enables a geometric interpretation of NMF solutions, clarifying equivalence classes of factorizations under permutation and orthogonal transformations. An intriguing numerical result, involving 400 NMF experiments across both real and synthetic datasets, show that in 99% of the cases, different algorithms tend to converge towards equivalent factor matrices. We benchmark eNMF against 9 state-of-the-art NMF algorithms with 9 initialization schemes across 3 real-world and 2 synthetic datasets. eNMF consistently outperforms all 81 competitors, achieving up to 30% lower reconstruction error under equal-time settings and up to 150% speedup under equal-error settings. The downstream experiments further demonstrate substantial performance gains in audio processing and recommendation tasks, corroborating the practical benefits

of the proposed exterior optimization framework. Code is available at `https://github.com/roychowdhuryresearch/eNMF`

## 1. Introduction and Motivation

Extracting low-dimensional representations of data is central to many analysis tasks. Classical approaches include Principal Component Analysis (PCA) (Abdi & Williams, 2010), Independent Component Analysis (ICA) (Hyvärinen & Oja, 2000; Boscolo et al., 2004), and Singular Value Decomposition (SVD). In cases where additional structure improves interpretability, constraints are imposed—for example, Network Component Analysis (NCA) (Liao et al., 2003; Boscolo et al., 2005) fixes zero entries using *a priori* knowledge. Nonnegative matrix factorization (NMF) approximates a nonnegative matrix as a low-rank product of two nonnegative matrices, yielding sparse, interpretable factors. NMF has been successfully applied across diverse domains, including hyperspectral imaging (Gillis, 2017), topic modeling (Arora et al., 2012b), audio denoising (Schmidt et al., 2007), video structuring (Essid & Févotte, 2013; Berry et al., 2007).

Recent advances in deep learning (DL) have produced generative models such as Variational Auto Encoders (VAEs) (Kingma & Welling, 2013) and Generative Adversarial Networks (GANs) (Goodfellow et al., 2014), whose latent spaces, like those in NMF, often correspond to interpretable features (Higgins et al., 2017; Bengio et al., 2013). Despite such progress, NMF remains of considerable interest both as a standalone tool and when integrated into modern frameworks. For example, nonnegativity in contrastive learning improves interpretability and disentanglement (Wang et al., 2024), while sparsity constraints in probabilistic matrix factorization enhance feature selection (Chang et al., 2021). NMF-style methods also support clustering with statistically optimal $K$-means guarantees (Zhuang et al., 2024), and reinforcement learning, where task decompositions enable reusable subtasks (Earle et al., 2018).

As reviewed in Section 2 and as demonstrated by our numerical results *our thesis is that the basic optimization problem of determining non-negative factors needs revisiting*. In

---

[*]Equal contribution  [1]ECE, UCLA [2]eBay Search Science Team. Correspondence to: Vwani Roychowdhury <vwani@g.ucla.edu>.

*Proceedings of the 43rd International Conference on Machine Learning*, Seoul, South Korea. PMLR 306, 2026. Copyright 2026 by the author(s).

particular, we explore the well-known rotational symmetry of the NMF factors. Let $U, V$ be a local minimum of the optimization problem: $\underset{U \geq 0, V \geq 0}{\text{minimize}} \ f(U, V) = \mathcal{G}(X - UV^\top)$, where $\mathcal{G}(\cdot)$ is a suitable norm, such as the Frobenius norm or the generalized Kullback–Leibler divergence (KLD). Then the manifold, $\mathcal{Y} = \{(UR, VR) \mid R^\top R = I, \ R \in \mathbb{R}^{r \times r}\}$, represents a level set: for any $(U_i, V_i) \in \mathcal{Y}$, $f(U_i, V_i) = f(U, V)$. Of course, such $(U_i, V_i) \in \mathcal{Y}$ are not guaranteed to be non-negative. But this observation when viewed in reverse (i.e. when the manifold is rooted at a point outside the positive orthant) forms the basis of our work, leading to a new framework for NMF and the following numerical results:

**1. From unconstrained optima to** locally optimal **NMF solutions.** Although the NMF problem is NP-complete (see Appendix A), the corresponding *unconstrained low-rank approximation* admits computationally efficient algorithms (e.g., via truncated SVD for the Frobenius norm case). We exploit the rotational invariance of the unconstrained optimum $(U^\star, V^\star)$ and explore the associated orthogonal manifold $\mathcal{Y}^*$. If this manifold $\mathcal{Y}^*$ intersects the positive orthant, the resulting factors are globally optimal for NMF as well. In general, however, such an intersection does not occur, and identifying an orthogonal transform $R$ that yields non-negative factors is itself NP-complete (see Section 3 and Appendix A). To address this challenge, we propose an ADMM-based heuristic that reaches a point closest to the positive orthant from the outside (Section 4). Starting from the resulting exterior point, we first enforce feasibility (see Section 4.2) and then use HALS (Cichocki & Phan, 2009) to converge to a local NMF minimum. We refer to this exterior-to-interior procedure as *eNMF*.

**2. Superior reconstruction accuracy and convergence speed.** As described in Sec. 5.2 *we pick 81 baseline algorithms*: nine algorithmic frameworks, and for each nine different initialization schemes to account for non-convexity. Our numerical results fall into *two classes* that highlight the geometry of the NMF optimization landscape. In the first class, for a standard parametrized dense synthetic benchmark, and several cases of a real-world Audio dataset (see Sec. 5.1), the SVD-seeded manifold $\mathcal{Y}^*$ *directly intersects* the positive orthant, allowing eNMF to reach the global NMF optimum via rotations alone. In contrast, leading interior-only methods show initial fast convergence but soon hit a barren plateau (Eckstein & Yao, 2015; Cerezo et al., 2021; Miao & Barthel, 2024) (see Fig. 1). Under equal-time budgets, competing methods can incur up to **25%** higher reconstruction error(see Fig. 2). In the *second class*, comprising three real datasets spanning text, images, and audio domains, while $\mathcal{Y}^*$'s do not intersect the postive orthant, the eNMF algorithm consistently reaches a local minimum (Appendix Table 12); the competing methods stall on barren plateaus and only asymptotically approach eNMF's re-

construction error. Matching eNMF's accuracy requires up to **500%** longer runtime for the next-best algorithms ( Tables 1, 2), demonstrating that *approaching the positive orthant from the exterior yields* both lower error and faster convergence.

**3. Geometric equivalence of solutions across algorithms.** We conduct a large-scale comparison of the factors produced by different NMF algorithms. Across more than 400 experimental settings, *distinct local minima occur in only four cases*; in all others, competing methods recover factors that are either permuted-and-scaled or rotated versions of eNMF's solution when reconstruction errors are comparable. This provides empirical evidence that many NMF solvers converge toward the same solution geometry, though at markedly different rates. To our knowledge, this is the first systematic study of factor equivalence across NMF algorithms.

**4. Improved downstream task performance.** Beyond reconstruction quality, we show that eNMF-derived factors translate into substantial downstream gains (see Appendix E). Across audio, vision, and recommendation tasks, eNMF yields over **10%** performance improvements in audio and vision benchmarks and over **50%** improvement in a top-$k$ recommendation task. These results demonstrate that improved optimization directly benefits practical applications.

## 2. Problem Statement and Prior work

A standard optimization framework for computing the non-negative factors uses the Frobenius norm

$$\underset{U \geq 0, V \geq 0}{\text{minimize}} \ f(U, V) = \tfrac{1}{2}\|X - UV^\top\|_F^2, \qquad (1)$$

where $X \geq 0$. There exist other variants of NMF depending on the distance measure used to quantify the quality of the approximation. For example, (Lee & Seung, 2000) propose both the Frobenius norm (as in (1)) and the generalized Kullback–Leibler divergence to evaluate approximation error; (Févotte et al., 2009) use the Itakura–Saito divergence for music analysis; and (Zhang et al., 2008) design a total variation norm to extract image patterns. In yet another variation (see Appendix C), the approximation-error minimization in (1) is performed only over a subset of the entries of the data matrix $X$. In such a case, it is referred to as the *matrix-completion factorization problem*, which we also address in Appendix C. The eNMF algorithm developed here is evaluated under the Frobenius-norm formulation but applies to all of these objective functions; it only requires an optimal solution to the unconstrained problem as the starting point.

Several algorithms have been developed to find good local minima of (1), generally falling into three classes: multiplicative updates, gradient-based methods, and alternating optimization / NNLS-based solvers. The prototypi-

cal multiplicative algorithm (Mult) (Lee & Seung, 2000) maintains nonnegativity throughout iterations but often converges slowly; accelerated variants such as A-HALS (Gillis & Glineur, 2012) and fast proximal/accelerated schemes (FPGM) (Xu, 2011) have therefore been proposed. The second class comprises gradient-based approaches, including projected/gradient multiplicative updates (Grad-Mult) (Lin, 2007a) and Nesterov-accelerated formulations (NeNMF) (Guan et al., 2012a), which take steps along the negative gradient and enforce nonnegativity via projection or proximal operators. The third class is based on alternating optimization, exploiting the convexity of (1) in either $U$ or $V$ so that updates reduce to solving non-negative least squares (NNLS). Representative variants include: ALS (Lin, 2007b) (unconstrained least squares + projection), AO-ADMM (Huang et al., 2016) (ADMM-based NNLS subproblems with efficient linear algebra), NMF-ADMM (Hajinezhad et al., 2016) (ADMM updates for the overall NMF objective), and HALS (Cichocki & Phan, 2009) (column/row-wise updates with projection), as well as rank-one downdating methods such as Vavasis R1D (Biggs et al., 2008). Despite their differences, these approaches all ensure feasibility by starting from nonnegative factors and/or enforcing nonnegativity after each iteration.

## 3. Exact NMF and Complexity of the NMF Problem

A special case of the NMF problem (1) is the *Exact NMF* problem, which takes as input a matrix $X \in \mathbb{R}^{n \times m}$ with $X \geq 0$ and rank $r \geq 1$, and asks whether there exist matrices $U \in \mathbb{R}^{n \times r}$ and $V \in \mathbb{R}^{m \times r}$, with $U, V \geq 0$, such that $X = UV^\top$. The complexity of the *Exact NMF* problem has been shown to be NP-hard (Vavasis, 2009). Since the exact factorization scenario is a special case of the general *NMF* problem, it follows that the *NMF* problem (1) is also NP-hard, and there is no provably efficient algorithm unless $P = NP$. Exact NMF is particularly relevant for the eNMF framework, as it is critical to guiding its geometric intuition. Moreover, if we synthetically construct $X$ from known nonnegative factors, then globally optimal ground-truth solutions are known, against which all NMF algorithms can be evaluated; see Fig. 1 and Appendix Table 13 for numerical examples.

We note that if $X$ admits an exact NMF factorization, then its SVD factors (outside the nonnegative orthant) can be mapped back to the NMF factors using linear transformations: As shown in (Vavasis, 2009), since $U^*V^{*T} = U_B V_B^T$ — where $(U^*, V^*)$ and $(U_B, V_B) \geq 0$ are the SVD and exact-NMF factors, respectively,— there must exist a matrix $Z \in \mathbb{R}^{r \times r}$ such that $U^* = U_A Z$ and $V^* = V_A Z^{-\top}$. Using this result, (Vavasis, 2009) proposed an intuitive rank-one-updates-based algorithmic strategy to iteratively estimate $Z$

and construct a $Q$, so that $(U^*Q, V^*Q^{-\top})$ can be close to the positive orthant and then feasible nonnegative factors can be obtained via projections. This intuitive idea, however, was not evaluated and later Vavasis *et al.* (Biggs et al., 2008) proposed the rank-one downdating method (R1D). R1D does not use the general rank-$r$ SVD factors $(U^*, V^*)$; however, it iteratively computes only rank-1 SVD factors of a non-negative matrix, which are themselves guaranteed to be non-negative. Thus, this ingenious R1D method works from inside the positive orthant like the rest of the NMF algorithms, and we use it as one of the nine baselines (Sec. 5.2). In Appendix A we, however, show how the above linear-transformation matrix $Z$ can be estimated using a sequence of rotation and scaling operations to solve the NMF problem and extend the results in the following section which restricts $Z$ to be only a rotation matrix.

## 4. Exterior NMF Algorithm

We start with a rank-$r$ truncated singular value decomposition (SVD) of the data matrix $X = U\Sigma V^\top$, $(U^\star = U\Sigma^{\frac{1}{2}}, V^\star = V\Sigma^{\frac{1}{2}})$. We consider the manifold:

$$\mathcal{Y}^\star = \{(U^\star R, V^\star R) \mid R^\top R = I, \; R \in \mathbb{R}^{r \times r}\}. \quad (2)$$

Our goal is to leverage the geometric framework introduced in Sec. 3, and compute an $R$ (as a restricted version of $Z$) such that the resulting factors $(U^\star R, V^\star R)$ are closest to the positive orthant. The proposed algorithm thus consists of three blocks: (i) **Optimal orthogonal transformation of the unconstrained global optimum** (Section 4.1): find an orthogonal transformation that moves the unconstrained global optimum close to the positive orthant. For several datasets, the transformed factors are nonnegative, thereby achieving globally optimal solutions. The next two blocks handle cases in which the transformed factors are infeasible. (ii) **Penalty method for attaining feasibility** (Section 4.2): attain feasibility of the transformed unconstrained global optimum using projected block coordinate descent. (iii) **Descent to a local minimum** (Section 4.3): descend the feasible factors to a local minimum of the NMF problem using hierarchical alternating least squares.

### 4.1. Optimal orthogonal transformation of unconstrained global optimum

An orthogonal transformation that minimizes the sum of the negative entries in $U^\star R$ and $V^\star R$ can be found by solving the optimization problem in (16) with $R_1 = \Lambda = I$. Although the problem is nonconvex, we can use ADMM as a heuristic to solve it (Boyd et al., 2011; Wang et al., 2019; Hong et al., 2016; Gao et al., 2020). To aid the ADMM formulation, introduce

$$W = \begin{bmatrix} U^\star \\ V^\star \end{bmatrix} \in \mathbb{R}^{(n+m) \times r}, \quad Z \in \mathbb{R}^{(n+m) \times r}, \quad (3)$$

where $Z$ is the splitting variable. With this notation, the problem can be written in standard ADMM form as

$$\underset{Z,R}{\text{minimize}} \quad \sum_{i=1}^{n+m} \sum_{j=1}^{r} h(Z_{ij})$$

$$\text{subject to} \quad Z - WR = 0,$$

$$R^\top R = \mathbf{I}. \tag{4}$$

The ADMM used to solve (4) is given in Algorithm 1.

---

**Algorithm 1** Optimal orthogonal transformation of unconstrained global optimum

---

1: **Input**: $U, V, R^{(0)}, Y^{(0)}, \rho$
2: Stack $U, V$ vertically to obtain $W$
3: **Repeat until convergence**:

4: $\quad Z^{(k+1)} \leftarrow \underset{Z}{\arg\min} \left( \sum_{i=1}^{n+m} \sum_{j=1}^{r} h(Z_{ij}) + \frac{\rho}{2} \|Z - WR^{(k)} + \frac{1}{\rho} Y^{(k)}\|_F^2 \right)$

5: $\quad R^{(k+1)} \leftarrow \underset{R^\top R = I}{\arg\min} \left( \frac{\rho}{2} \|Z^{(k+1)} - WR + \frac{1}{\rho} Y^{(k)}\|_F^2 \right)$

6: $\quad Y^{(k+1)} \leftarrow Y^{(k)} + \rho \left( Z^{(k+1)} - WR^{(k+1)} \right)$
7: $\quad k \leftarrow k + 1$
8: **Output**: $UR, VR$

---

#### 4.1.1. UPDATES FOR $Z$

We can update each entry of $Z$ independently. The updates are obtained by solving a simple quadratic problem:

$$Z_{ij}^{(k+1)} = \begin{cases} b_{ij}, & b_{ij} > 0, \\ 0, & -\frac{1}{\rho} \le b_{ij} \le 0, \\ b_{ij} + \frac{1}{\rho}, & b_{ij} < -\frac{1}{\rho}, \end{cases} \tag{5}$$

where $b_{ij} = \left( WR^{(k)} - \frac{1}{\rho} Y^{(k)} \right)_{ij}$.

#### 4.1.2. UPDATES FOR $R$

We can update $R$ by solving a nonconvex problem known as the orthogonal Procrustes problem (Gower & Dijksterhuis, 2004). If we define $B = Z^{(k+1)} + \frac{1}{\rho} Y^{(k)}$, then the update for $R$ is given by $R^{(k+1)} = CD^\top$, where $C$ and $D$ are obtained from the SVD $W^\top B = C \Sigma D^\top$.

Empirically, the ADMM rotation step is numerically stable across both synthetic and real datasets: the returned rotation matrices satisfy the orthogonality constraint to high precision and the procedure converges in only a few ADMM iterations (Appendix Tables 18–19).

### 4.2. Penalty method for attaining feasibility

The rotated point returned by Algorithm 1 is expected to be close to the positive orthant but not necessarily inside it (unless there exists an NMF solution that is also globally optimal for the unconstrained problem; see Fig. 1 (A) for an example). We use an exterior penalty method to attain feasibility of the NMF factors starting from the rotated point:

$$\underset{U,V}{\text{minimize}} \quad \frac{1}{2} \|X - UV^\top\|_F^2$$

$$+ \delta_u \sum_{i,j} h(U_{ij}) + \delta_v \sum_{i,j} h(V_{ij}),$$

where $h(q) = \max\{0, -q\}$ and $\delta_u, \delta_v$ are penalty parameters. We solve this problem using gradient descent. For sufficiently large $\delta_u, \delta_v$, the gradients on negative entries are dominated by the penalty terms, yielding approximate updates

$$U_{ij} \leftarrow U_{ij} + \rho_u \delta_u, \quad \forall (i,j) \in I_-, \tag{6}$$

$$V_{ij} \leftarrow V_{ij} + \rho_v \delta_v, \quad \forall (i,j) \in I_-, \tag{7}$$

where $I_-$ denotes the set of negative entries and $\rho_u, \rho_v$ are fixed step sizes. Updating the nonnegative entries of $U$ and $V$ in any given row does not affect the partial derivatives for nonnegative entries in other rows, and we can leverage this to update the rows of $U$ and $V$ independently. The updates for all nonnegative entries in the $i^{\text{th}}$ row are

$$U_{ij} \leftarrow \left( U_{ij} - d_i^\star (\nabla_U f)_{ij} \right)_+, \quad \forall (i,j) \in I_+, \tag{8}$$

$$V_{ij} \leftarrow \left( V_{ij} - t_i^\star (\nabla_V f)_{ij} \right)_+, \quad \forall (i,j) \in I_+, \tag{9}$$

where $I_+$ denotes the set of nonnegative entries, $\nabla_U f = (UV^\top - X)V$, $\nabla_V f = (UV^\top - X)^\top U$, $d_i^\star$ and $t_i^\star$ are optimal step sizes, and the projection $(x)_+ = \max\{0, x\}$ enforces nonnegativity. The optimal step sizes are (see Appendix B for a derivation)

$$d_i^\star = \frac{\left\| \{M_{U_+} \circ \nabla_U f\}(i,:) \right\|^2}{\left\| \{M_{U_+} \circ \nabla_U f\}(i,:) V^{(k)\top} \right\|^2}, \tag{10}$$

$$t_i^\star = \frac{\left\| \{M_{V_+} \circ \nabla_V f\}(i,:) \right\|^2}{\left\| \{M_{V_+} \circ \nabla_V f\}(i,:) U^{(k+1)\top} \right\|^2}. \tag{11}$$

We refer to this feasibility-attaining phase as the *ascent stage* because it yields a reconstruction error higher than that of the rotated SVD factors. Ablations starting from the same rotated SVD initialization show that the proposed feasibility-attainment stage followed by HALS descent is consistently faster than direct projection followed by descent under equal-error matching, and also achieves lower reconstruction error under identical wall-clock budgets; moreover, the closed-form row-wise step sizes in Eqs. (10)–(11) reduce runtime relative to fixed-step feasibility updates (Appendix

Tables 20, 25, and 21). The pseudocode for updating the nonnegative entries is given below.

---

**Algorithm 2** Projected Block Coordinate Descent (PBCD)

---
1: **Input**: $X, U, V, \epsilon, \texttt{max\_iter}, M_{U_+}, M_E$
2: $\nabla_{U_{\text{init}}} f \leftarrow M_{U_+} \circ \left( (M_E \circ (UV^\top - X))V \right)$
3: $n \leftarrow$ number of rows in $U$
4: **Repeat until** projected_grad $< \epsilon \times \|\nabla_{U_{\text{init}}} f\|_F$ **or** $\texttt{max\_iter}$:
5:    **for** $i = 1, \cdots, n$
6:       $d_i^\star \leftarrow$ using Eq. 10
7:       $U_{i,:} \leftarrow$ using Eq. 8
8:    projected_grad $\leftarrow \|M_{U_+} \circ \left( (M_E \circ (UV^\top - X))V \right)\|_F$
9: **Output**: $U$

---

**Algorithm 3** eNMF

---
1: **Input**: $X, r, \rho, \epsilon, \texttt{max\_iter}$
2: $U^\star, V^\star \leftarrow \text{SVD}(X, r)$
3: $(U^\star R, V^\star R) \leftarrow \text{orthogonal}(U^\star, V^\star, R^0, Y^0, \rho)$
4: $(U, V) \leftarrow (U^\star R, V^\star R)$
5: **Repeat until** $U, V$ **are nonnegative**:
6:    Update negative elements in $U$ using Eq. 6
7:    Compute mask $M_{U_+}$
8:    $U \leftarrow \text{PBCD}(X, U, V, \epsilon, \texttt{max\_iter}, M_{U_+}, \mathbf{1})$
9:    Update negative elements in $V$ using Eq. 7
10:   Compute mask $M_{V_+}$
11:   $V \leftarrow \text{PBCD}(X^\top, V, U, \epsilon, \texttt{max\_iter}, M_{V_+}, \mathbf{1})$
12: $U, V \leftarrow \text{HALS}(X, U, V)$
13: **Output**: $U, V$
14: **Note**: $\mathbf{1}$ is the matrix of all ones.

---

### 4.3. Descent to the local minimum

The ascent stage described in Section 4.2 returns factors that are already very close to a local minimum (as assessed by the KKT conditions), and any standard descent algorithm will move the feasible factors to a local minimum of the NMF problem. The PBCD algorithm outlined in the previous section is sufficient to move the feasible factors to a local minimum in reasonable time, but we use hierarchical alternating least squares (HALS) for the descent stage. The descent stage terminates once the factor matrices $U$ and $V$ converge to a local minimum, verified by checking the KKT optimality conditions for (1). The pseudocode for eNMF is provided below, consisting of the three main blocks introduced earlier:

In practice, the truncated-SVD initialization can be replaced by randomized SVD with nearly identical end-to-end run-time on real datasets, suggesting that eNMF depends primarily on the quality of the low-rank subspace rather than exact SVD computation (Appendix Table 23).

## 5. Experiments and Results

### 5.1. Datasets

**Exact factorization dataset:** We created a synthetic dataset that admits an exact factorization. The dataset was generated as follows: **(i)** sample the entries of the ground-truth factor matrices $U$ and $V$ from $U(0,1)$, **(ii)** apply a binary mask with sparsity $s$ to the ground-truth factors, and **(iii)** construct

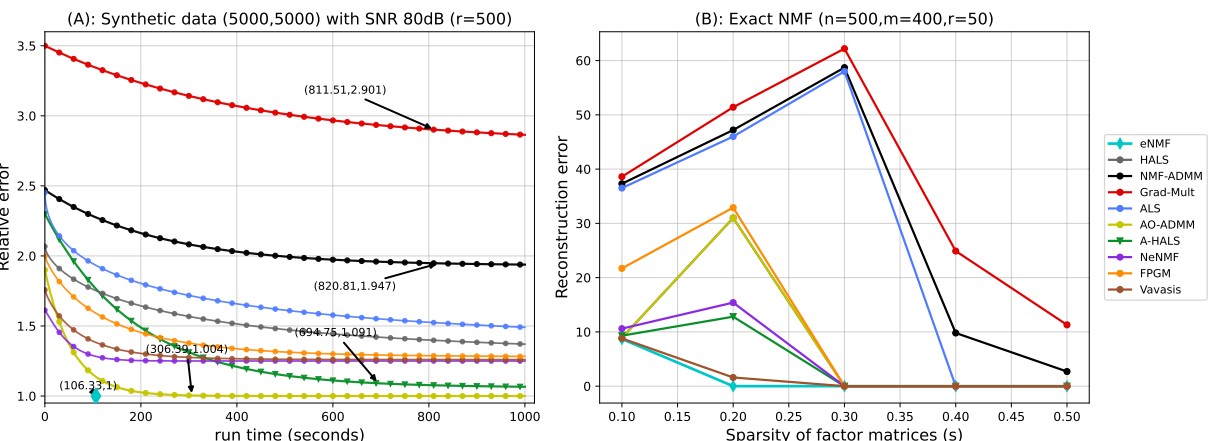

*Figure 1.* **(A) Given enough run time, all algorithms converge to local minima with similar reconstruction error**: Relative error $= \frac{\|X - U_{\text{NMF}} V_{\text{NMF}}^\top\|_F}{\|X - U_{\text{SVD}} V_{\text{SVD}}^\top\|_F}$ is plotted as a function of run time (seconds). *eNMF achieves the minimum (i.e., the unconstrained global minimum in this case) in 106 seconds, compared to hours of run time for some competing algorithms.* **(B) Exact factorization**: eNMF achieves the global minimum in the exact factorization dataset for sparsity levels 0.2 through 0.5. For a detailed reconstruction error for all algorithms please refer to Appendix Table. 13 **Note**: For each competitor of eNMF, results correspond to it's *best* run over a 9-initialization sweep (5 random initializations plus heuristic, K-means, NICA, and NNDSVD initializations).

the data matrix using $X = UV^\top$.

**Synthetic dataset** (*Dense and High-Dimensional $X$*): We generated $X$ as follows: **(i)** sample the entries of two matrices $W \in \mathbb{R}^{5,000 \times 1,000}$ and $H \in \mathbb{R}^{1,000 \times 5,000}$ from $U(0,1)$, **(ii)** sample the entries of $N \in \mathbb{R}^{5,000 \times 5,000}$ from $\mathcal{N}(0, \tau^2)$, where $\tau$ controls the signal-to-noise ratio (SNR), and **(iii)** form the data matrix $X = WH + N$.

**Verb dataset** (*Highly Sparse and Moderate-Dimensional $X$*): This dataset consists of entity–verb relationships extracted from a corpus of text data (Tangherlini et al., 2016), where $X(i,j)$ is the number of times entity $i$ appears as the object of verb $j$. Here $X \in \mathbb{R}^{282 \times 1,528}$ with 97.3% sparsity.

**Audio dataset**(*High-dimensional with temporal–spectral structure*): The dataset consists of English digit recordings (0–9) from 60 speakers. The resulting data matrix of size $24,000 \times 8,000$ is formed from spectrogram representations, where each row corresponds to the time–frequency pattern of a single utterance induced by speech articulation.

**Face dataset** (*Highly Skewed and High-Dimensional $X$*): The data matrix of shape $32,256 \times 64$ contains pixel values of a person's face under various illumination levels (Lee et al., 2005). Please see Appendix E for downstream tasks.

## 5.2. Baseline Algorithms

In addition to *eNMF*, we compared against several well-known *NMF* solvers: gradient-based multiplicative updates (Grad-Mult) (Lin, 2007a), alternating nonnegative least squares (ALS) (Lin, 2007b), alternating optimization with ADMM (AO-ADMM) (Huang et al., 2016), hierarchical alternating least squares (HALS) (Cichocki & Phan, 2009), and ADMM-based NMF (NMF-ADMM) (Hajinezhad et al., 2016). We further included accelerated HALS (A-HALS) (Gillis & Glineur, 2012), Nesterov-accelerated NMF (NeNMF) (Guan et al., 2012a), a fast proximal-gradient method (FPGM) (Xu, 2011), and the rank-one downdating (R1D) method of Vavasis *et al.* (Biggs et al., 2008). For each competing algorithm (all baselines except eNMF), we performed a multi-start evaluation with 9 runs following common NMF initialization strategies (Fathi Hafshejani & Moaberfard, 2022): 5 runs with random initialization (Lee & Seung, 1999), and 1 run each with clustering-based (K-means) initialization (Xue et al., 2008), heuristic initialization (Dong et al., 2014), NICA initialization (Kitamura & Ono, 2016), and NNDSVD initialization (Boutsidis & Gallopoulos, 2008). We report comparisons between eNMF and the *best* (highest-quality) run of each competing algorithm under this initialization sweep. We additionally compare against ANLS-BPP, a strong NNLS-based NMF solver, in Appendix Table 22; eNMF remains faster across all tested real-data settings under the equal-error protocol.

## 5.3. Performance comparison metrics: reconstruction error and run time

We observed an interesting phenomenon: if we run the competing algorithms long enough (often days), they achieve a reconstruction error similar to that of eNMF (Fig. 1). Therefore, for a meaningful comparison of relative performance, we conduct two experiments: (i) **Comparison of equal-time constrained reconstruction errors** (see Fig. 2 and 3; Appendix Tables 3 and 4): for any dataset and choice of $r$, we first run eNMF until it reaches a local minimum (verified via KKT conditions; see Appendix Table 12), record the resulting reconstruction error, and note the runtime. We then run the competing algorithms for at least as long as eNMF's runtime, thus giving them a fair chance to lower their error, by interrupting their iterations immediately after their execution time exceeds that of eNMF. If a competing algorithm reaches a local minimum (verified via KKT conditions) before the allotted equal-time budget, we terminate it early and record its runtime and final reconstruction error. (ii) **Comparison of equal-error constrained time complexity** (see Tables 1 and 2): we set the reconstruction-error target to that of eNMF and run the competing algorithms until they reach eNMF's reconstruction error. If a competing algorithm reaches a local minimum (verified via KKT conditions) before attaining eNMF's reconstruction error, we terminate it and report the reconstruction error at convergence. Moreover, for some datasets the optimization landscape may cause an algorithm to become trapped in a shallow valley such that the reconstruction error does not change for a large number of iterations; in such cases, if the reconstruction error fails to improve for 1,000 consecutive iterations (indicative of stagnation in a flat valley of the objective), we terminate the run and record the run time at the point of stagnation.

### 5.3.1. PERFORMANCE OF ENMF ON SYNTHETIC DATASETS

Across all SNR levels (20-100 dB) and latent dimensions ($r = 50$-$500$), eNMF consistently attains the *unconstrained global minimum* of the underlying low-rank factorization objective within the equal-time budget (Fig. 2). This behavior supports the central geometric intuition behind eNMF: the rotationally invariant manifold of equivalent unconstrained optima (anchored at the SVD solution) intersects the positive orthant, and a single, targeted *rotation step* is sufficient to reach a globally optimal nonnegative factorization in these regimes. Because eNMF reaches the minimum primarily through this rotation, rather than requiring prolonged constrained descent, it converges substantially faster than competing solvers. A phase-wise runtime decomposition confirms this behavior: on all synthetic settings, the feasibility/descent stage requires zero additional time, so the total runtime is fully accounted for by truncated-SVD ini-

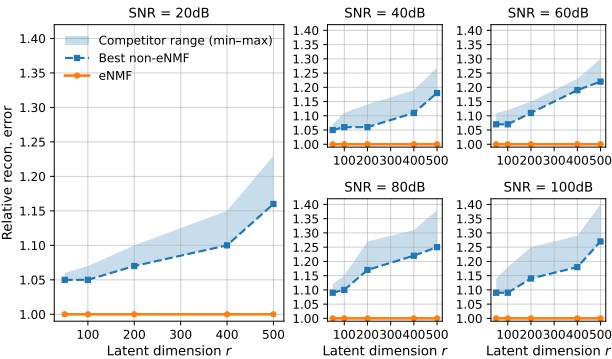

*Figure 2.* On synthetic datasets, Relative Reconstruction Error (RE) $\frac{\|X - U_{\text{NMF}} V_{\text{NMF}}^T\|_F}{\|X - U_{\text{SVD}} V_{\text{SVD}}^T\|_F}$ is plotted as a function of number of latent dimensions ($r$) for various SNR levels under identical wall-clock budgets. Shaded bands show the min–max range across competing methods (excluding eNMF) at each latent dimension ($r$); dashed lines indicate the best non-eNMF competitor. *eNMF achieves the unconstrained global minimum for all latent dimensions and across all SNR levels*. For each competitor, results correspond to its *best* run over a 9-initialization sweep (5 random initializations plus heuristic, K-means, NICA, and NNDSVD initializations). **Note**: For a detailed RE for all algorithms please refer to Appendix Table. 3.

tialization and ADMM-based rotation (Appendix Table 16). This advantage is most evident in the equal-error experiments (Table 1), where eNMF reaches the target error orders of magnitude sooner than all baselines across every (SNR, $r$) pair (e.g., typically 5–6× faster than AO-ADMM and 16–39× faster than HALS/NeNMF/FPGM/Vavasis and NMF-ADMM/Grad-Mult).

5.3.2. PERFORMANCE OF eNMF ON REAL DATASETS

Real datasets exhibit a markedly more nonconvex landscape than the synthetic setting, and the SVD-aligned rotationally invariant manifold does *not* necessarily intersect the positive orthant for every dataset and rank. Consistent with this geometry, we observe that **only for the Audio dataset at higher latent dimensions** ($r \in \{40, 80, 100\}$), the manifold intersects the positive orthant and eNMF reaches the *unconstrained global minimum* essentially via the rotation step (Fig. 3). This "one-shot" access to the optimum translates into substantially faster convergence compared to competing constrained solvers (Table 2).

For the remaining datasets/ranks where such an intersection is absent, eNMF still delivers the best performance under *both* evaluation protocols. Under **equal-time constrained** comparisons, eNMF consistently achieves the lowest reconstruction error across Face, Verb, and Audio, with reductions over the strongest baseline of up to ∼**8.4%** on Face (e.g., $r = 20$), ∼**5.4–7.4%** on Verb, and ∼**1.4–5.2%** on Audio (Fig. 3). Under the **equal-error constrained** protocol, eNMF reaches the target error fastest for every dataset

and rank, improving the runtime over the closest competitor by roughly ∼**14–16%** on Face, ∼**20–23%** on Verb, and ∼**22–26%** on Audio (Table 2).

We further evaluate scalability on an ultra-large sparse Reuters matrix of size $804{,}414 \times 47{,}236$ with $99.84\%$ sparsity, where eNMF remains faster than ANLS-BPP, HALS, and A-HALS across all tested latent dimensions under the equal-error protocol (Appendix Table 24).

We attribute eNMF's robustness on real data, even when the global-minimum intersection is absent, to the fact that its rotation step provides an *exceptionally strong warm start* near a low-error region of the feasible set, thereby avoiding long periods of slow constrained descent or stagnation in shallow valleys that can affect multiplicative-update and ADMM-style baselines. A phase-wise runtime decomposition on real datasets further shows that the feasibility/descent stage is lightweight relative to the SVD initialization and ADMM-based rotation stages, and vanishes entirely for Audio at $r \in \{40, 80, 100\}$ where the rotated SVD solution already reaches the target error (Appendix Table 17).As a result, eNMF reaches high-quality local minima faster and more reliably than competing algorithms across heterogeneous real-world regimes.

### 5.4. Equivalent local minima and relationships among factors obtained by different algorithms

Let $A$ and $B$ be two NMF algorithms that reach their respective local minima with factor matrices $(U_A, V_A)$ and $(U_B, V_B)$. We define *algorithms A and B as reaching equivalent local minima* if there exist a diagonal scaling matrix $\Sigma$ and an orthogonal matrix $R$ (with $R^\top R = I$) such that

$$\begin{bmatrix} U_A \\ V_A \end{bmatrix} = \begin{bmatrix} U_B \Sigma R \\ V_B \Sigma^{-1} R \end{bmatrix}.$$

*That is, $(U_B, V_B)$ are scaled and rotated versions of $(U_A, V_A)$, thereby preserving pairwise cosine similarity and clustering properties.* The cases of synthetic data and exact NMF are discussed in Appendices F and H, respectively. These show that the eNMF and AO-ADMM algorithms can lead to equivalent factors in certain cases with a general rotation matrix $R$ (see Appendix Table 11).

The more interesting case is where $R$ is a permutation matrix. Given that the competing algorithms tend to be at or near local minima, we ask whether their factors are trending toward the eNMF factors; that is, whether only a subset of the columns of $U_B$ and $V_B$ are permuted and scaled versions of the eNMF factors. Given two factor pairs, Appendix H provides an algorithm to determine such a set within a tolerance $\epsilon$ (we choose $\epsilon = 0.05$ here). To give competing algorithms the best chance, we ran all algorithms until they reached local minima or their reconstruction error plateaued (even when run for days). As listed in Appendix Table 12,

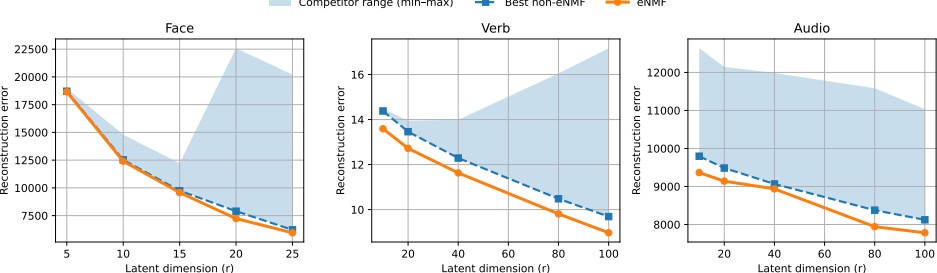

*Figure 3.* On real-world datasets, we plot absolute NMF reconstruction error versus latent dimension ($r$) under identical wall-clock budgets. The shaded envelope shows the min–max error across all competing methods (excluding eNMF) at each $r$, while dashed lines denote the best non-eNMF competitor. *eNMF consistently yields the lowest error across datasets and latent dimensions*. For each competitor, results correspond to its *best* run over a 9-initialization sweep (5 random intializations plus heuristic, K-means, NICA, and NNDSVD initializations). **Note**: For a detailed absolute NMF reconstruction error for all algorithms please refer to Appendix Table. 4.

*Table 1.* **Equal-error constrained time complexity on synthetic datasets.** For each competing algorithm, we report the runtime (in seconds) of its *best* run over a 9-initialization sweep (5 random initializations plus heuristic, K-means, NICA, and NNDSVD initializations), and compare against eNMF.

| SNR | $r$ | eNMF | HALS | NMF-ADMM | Grad-Mult | ALS | AO-ADMM | A-HALS | NeNMF | FPGM | Vavasis |
|---|---|---|---|---|---|---|---|---|---|---|---|
| 20dB | 50 | **176** | 3010.6 | 5113.0 | 6665.5 | 4586.5 | 966.2 | 1600.3 | 2716.7 | 2717.7 | 2745.5 |
| | 100 | **198.4** | 3393.7 | 5769.6 | 7513.4 | 5176.0 | 1098.2 | 1809.7 | 3079.3 | 3078.1 | 3103.6 |
| | 200 | **233.6** | 4038.3 | 6854.2 | 8919.9 | 6151.4 | 1300.6 | 2172.6 | 3689.4 | 3702.7 | 3735.3 |
| | 400 | **281.6** | 4904.1 | 8267.9 | 10912.6 | 7476.2 | 1620.7 | 2679.8 | 4538.8 | 4580.3 | 4559.8 |
| | 500 | **320** | 5595.8 | 9453.3 | 12512.8 | 8490.6 | 1864.9 | 3040.1 | 5265.2 | 5263.2 | 5298 |
| 40dB | 50 | **111.98** | 1898.8 | 3245.1 | 4176.8 | 2895.6 | 611.4 | 988.3 | 1704.2 | 1703.5 | 1706 |
| | 100 | **126.23** | 2143.5 | 3635.2 | 4704.6 | 3251.5 | 696.6 | 1140.6 | 1928.6 | 1928.3 | 1912.4 |
| | 200 | **148.63** | 2527.5 | 4263.3 | 5523.0 | 3841.7 | 826.5 | 1343.1 | 2279.8 | 2262.4 | 2258.9 |
| | 400 | **179.17** | 3120.0 | 5251.3 | 6821.0 | 4762.9 | 990.6 | 1595.1 | 2820.6 | 2855.4 | 2862.0 |
| | 500 | **203.6** | 3532.5 | 5996.2 | 7924.7 | 5512.0 | 1117.2 | 1834.4 | 3261.9 | 3206.5 | 3120.4 |
| 60dB | 50 | **81.83** | 1372.4 | 2345.0 | 3001.5 | 2121.0 | 449.2 | 724.3 | 1234.6 | 1236.1 | 1281.5 |
| | 100 | **92.24** | 1558.5 | 2664.5 | 3413.2 | 2416.9 | 509.3 | 831.6 | 1414.9 | 1411.4 | 1448.3 |
| | 200 | **108.61** | 1836.4 | 3121.0 | 3993.9 | 2842.3 | 600.0 | 985.5 | 1671.4 | 1689.0 | 1726.6 |
| | 400 | **130.93** | 2226.1 | 3764.3 | 4805.1 | 3433.4 | 719.7 | 1181.7 | 2011.8 | 2014.6 | 2050.6 |
| | 500 | **148.78** | 2528.5 | 4290.8 | 5493.7 | 3893.8 | 820.4 | 1344.8 | 2288.2 | 2302.1 | 2308.3 |
| 80dB | 50 | **58.48** | 985.6 | 1683.1 | 2142.3 | 1510.9 | 322.5 | 519.9 | 909.1 | 910.3 | 930.5 |
| | 100 | **65.92** | 1098.2 | 1881.1 | 2406.7 | 1683.9 | 350.5 | 565.4 | 981.2 | 981.6 | 977.4 |
| | 200 | **77.62** | 1303.8 | 2234.9 | 2877.4 | 1990.7 | 415.0 | 687.3 | 1184.7 | 1184.2 | 1161.3 |
| | 400 | **93.57** | 1600.0 | 2718.8 | 3498.7 | 2438.9 | 512.9 | 851.2 | 1442.3 | 1432.7 | 1464.6 |
| | 500 | **106.33** | 1812.0 | 3093.7 | 4043.9 | 2773.9 | 596.2 | 962.8 | 1679.8 | 1675.6 | 1673.6 |
| 100dB | 50 | **49.5** | 810.2 | 1409.0 | 1788.7 | 1255.5 | 258.2 | 415.7 | 730.7 | 717.9 | 718.9 |
| | 100 | **55.8** | 915.0 | 1593.4 | 2020.5 | 1414.6 | 291.9 | 467.8 | 820.5 | 816.9 | 818.0 |
| | 200 | **65.7** | 1081.7 | 1875.1 | 2381.6 | 1676.0 | 349.5 | 557.0 | 967.7 | 968.1 | 987.5 |
| | 400 | **79.2** | 1340.1 | 2277.5 | 2923.2 | 2052.4 | 429.9 | 689.5 | 1203.6 | 1208.1 | 1193.2 |
| | 500 | **90** | 1521.4 | 2596.5 | 3334.9 | 2329.0 | 487.8 | 798.8 | 1401.4 | 1397.4 | 1390.5 |

*Table 2.* **Equal-error constrained time complexity on real datasets.** For each competing algorithm, we report the runtime (in seconds) of its *best* run over a 9-initialization sweep (5 random initializations plus heuristic, K-means, NICA, and NNDSVD initializations), and compare against eNMF.

| Dataset | $r$ | eNMF | HALS | NMF-ADMM | Grad-Mult | ALS | AO-ADMM | A-HALS | NeNMF | FPGM | Vavasis |
|---|---|---|---|---|---|---|---|---|---|---|---|
| Face | 5 | **9.8** | 13.43 | 16.07 | 17.35 | 14.80 | 11.53 | 12.06 | 12.74 | 13.46 | 14.65 |
| | 10 | **132.7** | 181.79 | 219.95 | 232.78 | 193.70 | 157.27 | 163.52 | 170.81 | 185.76 | 199.11 |
| | 15 | **203.1** | 280.28 | 335.12 | 359.52 | 296.60 | 235.77 | 253.04 | 257.91 | 282.34 | 302.59 |
| | 20 | **246.4** | 333.19 | 396.70 | 426.60 | 364.83 | 291.47 | 307.99 | 318.11 | 339.04 | 370.06 |
| | 25 | **311.8** | 430.48 | 502.72 | 553.36 | 462.03 | 360.61 | 392.17 | 400.90 | 429.85 | 463.77 |
| Verb | 10 | **5.17** | 8.32 | 8.74 | 9.72 | 8.19 | 6.57 | 7.24 | 7.57 | 8.01 | 8.12 |
| | 20 | **4.93** | 7.49 | 8.16 | 8.90 | 7.59 | 6.38 | 6.73 | 6.93 | 7.42 | 7.86 |
| | 40 | **8.76** | 14.19 | 14.89 | 16.64 | 13.23 | 10.99 | 12.42 | 12.61 | 13.93 | 13.45 |
| | 80 | **12.28** | 19.89 | 20.47 | 22.41 | 19.03 | 15.96 | 16.85 | 17.34 | 18.67 | 19.53 |
| | 100 | **14.66** | 23.16 | 24.63 | 27.41 | 22.43 | 18.54 | 20.67 | 21.55 | 23.46 | 22.87 |
| Audio | 10 | **27.81** | 39.49 | 61.18 | 69.52 | 47.28 | 35.87 | 40.60 | 37.27 | 44.77 | 42.27 |
| | 20 | **54.43** | 86.54 | 103.96 | 119.75 | 87.09 | 75.11 | 73.48 | 80.01 | 80.56 | 84.37 |
| | 40 | **62.11** | 86.33 | 139.75 | 145.96 | 107.45 | 81.36 | 91.30 | 80.12 | 101.24 | 95.03 |
| | 80 | **70.29** | 108.95 | 144.09 | 160.26 | 113.87 | 98.41 | 95.59 | 101.92 | 105.44 | 108.25 |
| | 100 | **107.74** | 158.38 | 226.25 | 263.96 | 181.00 | 143.29 | 155.15 | 148.68 | 170.23 | 167.00 |

the KKT conditions show that our eNMF algorithm reaches a local minimum for all datasets and all latent dimensions. For permutation checking, we fix algorithm $A$ to be eNMF and compare its factors with those obtained from the competing algorithms. In Appendix Table 14, we report, as a function of the latent dimension for the Real datasets, the percentage of columns of the factor matrices that have converged to a unique column of the eNMF factors along with the error ratio. From the table, we make the following claim: *the factors from the NMF algorithms converge to a pair of factor matrices that are unique modulo permutation and scaling only at the local minima (error ratio = 1), and even a small deviation from the minima leads to a large reduction in the number of columns that have converged to a unique column of the eNMF.* Only in few instances (see Appendix Table 14) did the competing algorithms converge to a non-equivalent minimum with higher reconstruction error; that is, the factor matrices do not trend toward those of the eNMF algorithm, yet the KKT conditions are satisfied.

## 5.5. Downstream utility of eNMF factors under equal-time learning

Beyond improved reconstruction quality and faster convergence, eNMF also learns *highly transferable* low-rank representations that consistently benefit downstream tasks. Since downstream applications are time-sensitive, we evaluate all tasks using factor matrices learned under an **equal wall-clock budget** for every method, ensuring a fair comparison of representation quality in practical settings. Across three domains-vision (Face recognition), audio (AudioMNIST digit classification), and recommendation (MovieLens 1M)-eNMF-derived features outperform all competitors for every tested rank. On the Face recognition task, eNMF improves accuracy by **+5.7 to +10.3 percentage points** over the best baseline across ranks (Appendix Table 5). On AudioMNIST, eNMF yields even larger gains of **+8.5 to +12.5 points** (Appendix Table 6). For MovieLens 1M rating prediction, eNMF reduces RMSE by ∼**7–12%** relative to the strongest baseline (Appendix Table 9), while in top-$K$ recommenda-

tion it achieves substantial improvements of up to ~**100%** in NDCG@10 and ~**22–33%** in Recall@10 (Appendix Table 10). Together, these results show that eNMF not only excels as an optimizer, but also produces factors that are consistently more discriminative and effective for downstream learning across diverse modalities. Post-rotation ablations on AudioMNIST further show that different correction strategies yield comparable downstream accuracy once matched to the same reconstruction-error target, but under the same wall-clock budget the proposed feasibility-attainment plus HALS stage produces substantially better downstream features at lower ranks (Appendix Tables 7–8).

## 6. Concluding remarks

We developed a general framework for the *NMF* problem that performs significantly better than existing approaches when compared on both reconstruction error and run time. eNMF framework also enables us to explore the geometry of the NMF problem: (i) How the NMF factors relate to the unconstrained matrix factors; (ii) Different algorithms trend towards equivalent factors in most instances; and (iii) at least four examples of non-equivalent local minima with different reconstruction errors. However, NMF is used as a feature extraction tool, and the question remains whether eNMF's superior performance can be translated to significantly better performance. We present three case studies in the Appendix E: (i) face recognition task (ii) audio digit recognition task (iii) movie recommendation task. We find that eNMF can yield significant performance enhancements; for example, more than $10\%$ in face and audio recognition task and more than $50\%$ in a top-$k$ recommendation task.

Finally, we would like to note that there have been recent progress on designing Online NMF (oNMF) algorithms. oNMF updates factors incrementally as new data arrives, enabling scalability to very large or streaming datasets (Mairal et al., 2010; Guan et al., 2012b). These methods have been applied in domains such as topic modeling, audio separation, and recommender systems, but they primarily address efficiency and data-stream settings rather than the optimization geometry of NMF. Our work is complementary: eNMF focuses on the batch setting and shows that exterior initialization from unconstrained factors improves convergence and reconstruction quality compared to state-of-the-art interior algorithms. As part of future work it would be interesting to explore if eNMF methodologies can be extended to the online case to obtain better performance, paralleling what we have shown for the batch NMF case.

## Acknowledgments

We thank Professor Lieven Vandenberghe for valuable feedback and refining the algorithmic design of eNMF.

## Impact Statement

This work advances nonnegative matrix factorization by introducing an exterior optimization framework that improves runtime and reconstruction quality across several benchmark settings. Because NMF is widely used for interpretable representation learning in domains such as text analysis, audio processing, recommendation systems, and biomedical data analysis, faster and more reliable solvers may benefit both research and practical applications.

The proposed method is algorithmic in nature and does not introduce a new data-collection pipeline or autonomous decision-making system. However, like any representation-learning method, its impact depends on the downstream application and the data on which it is used. When applied to sensitive datasets, practitioners should consider privacy, bias, robustness, and appropriate domain validation. We do not identify direct negative societal impacts specific to the proposed algorithm beyond the general risks associated with applying matrix factorization methods in high-stakes settings.

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

# A. Exact NMF and the complexity of the NMF problem

A special case of the NMF problem (1) is the *Exact NMF* problem, which takes as input a matrix $X \geq 0 \in \mathbb{R}^{n \times m}$ with rank $r \geq 1$, and asks whether there exists a pair of matrices $(U, V)$, where $U \geq 0 \in \mathbb{R}^{n \times r}$ and $V \geq 0 \in \mathbb{R}^{m \times r}$, such that $X = UV^\top$.

The complexity of the *Exact NMF* problem has been studied extensively in the literature. (Vavasis, 2009) and (Arora et al., 2012a) showed that this problem is NP-hard and that there is no $(nm)^{o(r)}$-time algorithm for *Exact NMF*, unless $P = NP$. For some special cases of the *Exact NMF* problem, namely the separable case, (Donoho & Stodden, 2004) and later (Arora et al., 2012a) developed polynomial-time algorithms in $n, m$, and $r$ to compute the exact factorization, if one exists. Since the exact factorization scenario is a special case of the general *NMF* problem, it implies that the *NMF* problem (1) is also NP-hard, and we cannot find any provably efficient algorithm unless $P = NP$. To develop a geometric intuition let us consider the singular value decomposition (SVD) of the data matrix $X = U\Sigma V^\top$, $(U^\star = U\Sigma^{\frac{1}{2}}, V^\star = V\Sigma^{\frac{1}{2}})$. Thus we get $X = UV^\top = U^\star V^{\star\top}$, where $U, V$ are non-negative and the SVD factors are not necessarily so. As shown in (Vavasis, 2009), if $U_A V_A^T = U_B V_B^T$ there must exist a matrix $Z \in \mathbb{R}^{r \times r}$ such that $U_B = U_A Z$ and $V_B = V_A Z^{-\top}$. By considering an SVD of $Z$, $Z = R_1 \Lambda R_2^\top$, we find that there exist orthogonal matrices $R_1, R_2 \in \mathbb{R}^{r \times r}$ $(R_1 R_1^\top = I = R_2 R_2^\top)$ and a diagonal scaling matrix $\Lambda$ such that $U = U_A^\star R_1 \Lambda R_2^\top$, $V = V^{\star\top} R_1 \Lambda^{-1} R_2^\top$. That is, the non-negative factors $U, V$ can be obtained from $U_A^\star, V^{\star\top}$ by solving for two rotation matrices, and two diagonal scaling matrices.

## A.1. Going from SVD to known Exact NMF factors

Let $U_1, U_2 \in \mathbb{R}^{n \times r}$ and $V_1, V_2 \in \mathbb{R}^{m \times r}$ be matrices with full column rank and suppose $U_1 V_1^\top = U_2 V_2^\top$. Then there exists $A \in \mathbb{R}^{r \times r}$ such that $U_2 = U_1 A$ and $V_2 = V_1 A^{-\top}$ (Vavasis, 2009; Stewart, 2001).

In the context of Exact NMF, let

$$X = U_0 V_0^\top,$$

where $X \geq 0 \in \mathbb{R}^{n \times m}$ is a nonnegative data matrix and $U_0 \geq 0 \in \mathbb{R}^{n \times r}$, $V_0 \geq 0 \in \mathbb{R}^{m \times r}$ are the Exact NMF factors. Also, let $X$ admit the following SVD:

$$X = U\Sigma V^\top,$$

and construct the scaled SVD factor matrices as

$$U^\star = U\Sigma^{\frac{1}{2}},$$
$$V^\star = V\Sigma^{\frac{1}{2}}.$$

Since

$$U_0 V_0^\top = U^\star V^{\star\top},$$

and all these matrices have full column rank, by the result stated above there exists $A$ such that

$$U_0 = U^\star A,$$
$$V_0 = V^\star A^{-\top}. \tag{12}$$

If we can solve (12) for $A$, then we have a way to go from the SVD factors to known Exact NMF factors. This exercise (going from SVD to known Exact NMF factors) motivates the optimization problem for the inverse case, where we must solve for an $A$ that transforms SVD factors to *unknown* Exact NMF factors.

Let the SVD of $A$ be

$$A = R_1 \Lambda R_2^\top,$$

where $R_1, R_2 \in \mathbb{R}^{r \times r}$ are orthogonal matrices and $\Lambda$ is a diagonal scaling matrix. Substituting the SVD of $A$ into (12) gives

$$U_0 = U^\star R_1 \Lambda R_2^\top,$$
$$V_0 = V^\star R_1 \Lambda^{-1} R_2^\top. \tag{13}$$

Using $U^\star = U\Sigma^{1/2}$ in (13) yields

$$U_0 = U\Sigma^{\frac{1}{2}} R_1 \Lambda R_2^\top. \tag{14}$$

Multiplying both sides of (14) on the left by $\Sigma^{-\frac{1}{2}} U^\top$ gives

$$\Sigma^{-\frac{1}{2}} U^\top U_0 = R_1 \Lambda R_2^\top = A. \tag{15}$$

Hence, we obtain $A$ from the SVD of $\Sigma^{-\frac{1}{2}} U^\top U_0$.

### A.2. Going from SVD to unknown Exact NMF factors

To solve the Exact NMF problem, we can first compute the scaled SVD factors $(U^\star, V^\star)$ of a nonnegative $X$, and then solve for $R_1, R_2 \in \mathbb{R}^{r \times r}$ ($R_1 R_1^\top = I = R_2 R_2^\top$) and a diagonal scaling matrix $\Lambda$ such that $U^\star R_1 \Lambda R_2 \geq 0$ and $V^\star R_1 \Lambda^{-1} R_2 \geq 0$. If $X$ has an exact NMF solution, then this problem is feasible. One can define the following nonconvex optimization problem to solve for the orthogonal and scaling matrices:

$$\underset{R_1, R_2, \Lambda}{\text{minimize}} \quad \sum_{i=1}^{n} \sum_{j=1}^{r} h\big((U^\star R_1 \Lambda R_2)_{ij}\big)$$
$$+ \sum_{i=1}^{m} \sum_{j=1}^{r} h\big((V^\star R_1 \Lambda^{-1} R_2)_{ij}\big) \tag{16}$$
$$\text{subject to} \quad R_1^\top R_1 = I, \ \ R_2^\top R_2 = I, \ \ \Lambda \geq 0,$$

where $h(q) = \max\{0, -q\}$. *Clearly the optimization problem in* (16) *has minimum value* $= 0$ *if and only if $X$ admits an exact solution.* This, however, is an NP-hard problem, and our ADMM implementations for solving (16) (a generalized version of the eNMF algorithm presented in Section 4) yielded exact nonnegative solutions in reasonable time only when $X$ has low dimensions. For example, for synthetic datasets (see Section 5.1 for the data generation procedure) with dimensions $n = 50, m = 40, r = 10, s = 0.1$ and $n = 100, m = 80, r = 20, s = 0.1$ (where $s$ is the sparsity factor), the ADMM implementation yielded almost exact nonnegative solutions. However, for $X$ with large dimensions $n, m$, the ADMM implementations *yielded factors that are only close to the positive orthant,* **which are still exterior infeasible points** (see Fig. 4 for details). *This computational intractability of finding optimal $R_1, \Lambda, R_2$ to move the SVD solutions to the feasible region* (even when such solutions exist) *prompted a more approximate approach, as described in Section 4 in the context of the general NMF problem.*

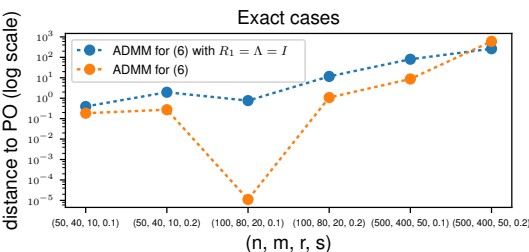

*Figure 4.* Distance to the positive orthant (as defined by the sum of the absolute values of the negative entries of the factor matrices post optimization; see Eqs. (16) and (4)) is plotted for synthetic exact datasets with various dimensions and sparsity levels. For lower dimensions, the ADMM implementation of (16) generates exact nonnegative solutions. For larger dimensions, ADMM implementations for both the general and relaxed versions (involving only a single orthogonal matrix $R_2$ and setting $R_1 = \Lambda = I$) generate factors that are only close to the positive orthant (PO), thus requiring further processing.

### A.3. ADMM implementation for solving (16)

We solve (16) using ADMM. To aid the ADMM formulation, we introduce the following notation:

$$W_{\text{svd}} = \begin{bmatrix} U^\star \\ V^\star \end{bmatrix} \in \mathbb{R}^{(n+m) \times r}, \qquad W_1, W_2, W_3,$$

where $W_1, W_2, W_3$ are splitting variables. Then, with the above notation we have the following equivalent optimization problem:

$$
\begin{aligned}
\underset{W_1, W_2, W_3, R_1, R_2}{\text{minimize}} \quad & \sum_{i=1}^{n+m} \sum_{j=1}^{r} h([W_3]_{ij}) \\
\text{subject to} \quad & W_1 = \begin{bmatrix} W_{11} \\ W_{12} \end{bmatrix} = \begin{bmatrix} U^\star \\ V^\star \end{bmatrix} R_1 = W_{\text{svd}} R_1, \\
& W_2 = \begin{bmatrix} W_{21} \\ W_{22} \end{bmatrix} = \begin{bmatrix} W_{11} \Lambda \\ W_{12} \Lambda^{-1} \end{bmatrix}, \\
& W_3 = W_2 R_2, \\
& R_1^\top R_1 = I, \\
& R_2^\top R_2 = I.
\end{aligned}
\tag{17}
$$

The ADMM for (17) is given by Algorithm 4.

---

**Algorithm 4** ADMM for (17)

---

**Input**: $U^\star, V^\star$
Stack $U^\star, V^\star$ vertically to get $W_{\text{svd}}$
**Repeat until convergence**:
$(W_1^{(k+1)}, W_3^{(k+1)}) \leftarrow \underset{W_1, W_3}{\arg\min} \, f_1(W_1, W_3)$
$(W_2^{(k+1)}, R_1^{(k+1)}) \leftarrow \underset{W_2, R_1}{\arg\min} \, f_2(W_2, R_1)$
$(R_2^{(k+1)}, \Lambda^{(k+1)}) \leftarrow \underset{R_2, \Lambda}{\arg\min} \, f_3(R_2, \Lambda)$
$M^{(k+1)} \leftarrow M^{(k)} + W_1^{(k+1)} - W_{\text{svd}} R_1^{(k+1)}$
$\begin{bmatrix} N_1 \\ N_2 \end{bmatrix}^{(k+1)} \leftarrow \begin{bmatrix} N_1 \\ N_2 \end{bmatrix}^{(k)} + \begin{bmatrix} W_{21} \\ W_{22} \end{bmatrix}^{(k+1)} - \begin{bmatrix} W_{11} \Lambda \\ W_{12} \Lambda^{-1} \end{bmatrix}^{(k+1)}$
$P^{(k+1)} \leftarrow P^{(k)} + W_3^{(k+1)} - W_2^{(k+1)} R_2^{(k+1)}$
$k \leftarrow k + 1$
**Output**: $R_1, R_2, \Lambda$

---

In Algorithm 4, we have three subproblems with the following cost functions:

$$
\begin{aligned}
f_1(W_1, W_3) = & \sum_{i=1}^{n+m} \sum_{j=1}^{r} h([W_3]_{ij}) + \frac{\rho}{2} \left\| W_1 - W_{\text{svd}} R_1^{(k)} + M^{(k)} \right\|_F^2 \\
& + \frac{\gamma}{2} \left\| W_{21}^{(k)} - W_{11} \Lambda^{(k)} + N_1^{(k)} \right\|_F^2 + \frac{\gamma}{2} \left\| W_{22}^{(k)} - W_{12} \Lambda^{(k)-1} + N_2^{(k)} \right\|_F^2 \\
& + \frac{t}{2} \left\| W_3 - W_2^{(k)} R_2^{(k)} + P^{(k)} \right\|_F^2, \\
f_2(W_2, R_1) = & \frac{\rho}{2} \left\| W_1^{(k+1)} - W_{\text{svd}} R_1 + M^{(k)} \right\|_F^2 + \frac{\gamma}{2} \left\| W_{21} - W_{11}^{(k+1)} \Lambda^{(k)} + N_1^{(k)} \right\|_F^2 \\
& + \frac{\gamma}{2} \left\| W_{22} - W_{12}^{(k+1)} \Lambda^{(k)-1} + N_2^{(k)} \right\|_F^2 + \frac{t}{2} \left\| W_3^{(k+1)} - W_2 R_2^{(k)} + P^{(k)} \right\|_F^2, \\
f_3(R_2, \Lambda) = & \frac{\gamma}{2} \left\| W_{21}^{(k+1)} - W_{11}^{(k+1)} \Lambda + N_1^{(k)} \right\|_F^2 + \frac{\gamma}{2} \left\| W_{22}^{(k+1)} - W_{12}^{(k+1)} \Lambda^{-1} + N_2^{(k)} \right\|_F^2 \\
& + \frac{t}{2} \left\| W_3^{(k+1)} - W_2^{(k+1)} R_2 + P^{(k)} \right\|_F^2.
\end{aligned}
$$

### A.3.1. UPDATES FOR $W_1$ AND $W_3$

$f_1(W_1, W_3)$ is separable in $W_1$ and $W_3$, so they can be updated in parallel.

**Updates for $W_3$:** We can update each entry of $W_3$ independently by solving

$$[W_3]_{ij}^{(k+1)} = \operatorname*{arg\,min}_{[W_3]_{ij}} h([W_3]_{ij}) + \frac{t}{2}\big([W_3]_{ij} - [W_2^{(k)}R_2^{(k)} - P^{(k)}]_{ij}\big)^2.$$

The updates are

$$[W_3]_{ij}^{(k+1)} = \begin{cases} b_{ij}, & b_{ij} > 0, \\ 0, & -\frac{1}{t} \le b_{ij} \le 0, \\ b_{ij} + \frac{1}{t}, & b_{ij} < -\frac{1}{t}, \end{cases}$$

where $b_{ij} = [W_2^{(k)}R_2^{(k)} - P^{(k)}]_{ij}$.

**Updates for $W_1$:** The blocks $W_{11}$ and $W_{12}$ are updated in parallel by solving

$$W_{11}^{(k+1)} = \operatorname*{arg\,min}_{W_{11}} \frac{\rho}{2}\big\|W_{11} - U^\star R_1^{(k)} + M_1^{(k)}\big\|_F^2 + \frac{\gamma}{2}\big\|W_{11}\Lambda^{(k)} - W_{21}^{(k)} - N_1^{(k)}\big\|_F^2,$$

$$W_{12}^{(k+1)} = \operatorname*{arg\,min}_{W_{12}} \frac{\rho}{2}\big\|W_{12} - V^\star R_1^{(k)} + M_2^{(k)}\big\|_F^2 + \frac{\gamma}{2}\big\|W_{12}\Lambda^{(k)} - W_{22}^{(k)} - N_2^{(k)}\big\|_F^2.$$

Elementwise closed forms:

$$[W_{11}]_{ij}^{(k+1)} = \frac{\rho\,[U^\star R_1^{(k)} - M_1^{(k)}]_{ij} + \gamma\,\lambda_j\,[W_{21}^{(k)} + N_1^{(k)}]_{ij}}{\rho + \gamma\lambda_j^2},$$

$$[W_{12}]_{ij}^{(k+1)} = \frac{\rho\,[V^\star R_1^{(k)} - M_2^{(k)}]_{ij} + \frac{\gamma}{\lambda_j}\,[W_{22}^{(k)} + N_2^{(k)}]_{ij}}{\rho + \frac{\gamma}{\lambda_j^2}},$$

where $\lambda_j$ is the $j$th diagonal element of $\Lambda$ (note: $\Lambda \in \mathbb{R}^{r \times r}$).

### A.3.2. UPDATES FOR $W_2$ AND $R_1$

$f_2(W_2, R_1)$ is separable in $W_2$ and $R_1$, so they can be updated in parallel.

**Updates for $R_1$:** We update $R_1$ by solving an orthogonal Procrustes problem. Let $B = W_1^{(k+1)} + M^{(k)}$; then

$$R_1^{(k+1)} = CD^\top, \tag{18}$$

where $C$ and $D$ are obtained from the SVD $W_{\text{svd}}^\top B = C\Sigma D^\top$.

**Updates for $W_2$:** The blocks $W_{21}$ and $W_{22}$ are updated in parallel via

$$W_{21}^{(k+1)} = \operatorname*{arg\,min}_{W_{21}} \frac{\gamma}{2}\big\|W_{21} - W_{11}^{(k+1)}\Lambda^{(k)} + N_1^{(k)}\big\|_F^2 + \frac{t}{2}\big\|W_{31}^{(k+1)} - W_{21}R_2^{(k)} + P^{(k)}\big\|_F^2,$$

$$W_{22}^{(k+1)} = \operatorname*{arg\,min}_{W_{22}} \frac{\gamma}{2}\big\|W_{22} - W_{12}^{(k+1)}\Lambda^{(k)-1} + N_2^{(k)}\big\|_F^2 + \frac{t}{2}\big\|W_{32}^{(k+1)} - W_{22}R_2^{(k)} + P_2^{(k)}\big\|_F^2.$$

Closed forms:

$$W_{21}^{(k+1)} = \frac{\gamma\,W_{11}^{(k+1)}\Lambda^{(k)} - \gamma\,N_1^{(k)} + t\big(W_{31}^{(k+1)} + P_1^{(k)}\big)R_2^{(k)\top}}{\gamma + t},$$

$$W_{22}^{(k+1)} = \frac{\gamma\,W_{12}^{(k+1)}\Lambda^{(k)-1} - \gamma\,N_2^{(k)} + t\big(W_{32}^{(k+1)} + P_2^{(k)}\big)R_2^{(k)\top}}{\gamma + t}.$$

### A.3.3. UPDATES FOR $R_2$ AND $\Lambda$

$f_3(R_2, \Lambda)$ is separable in $R_2$ and $\Lambda$, so they can be updated in parallel.

**Updates for** $R_2$: Update $R_2$ via an orthogonal Procrustes problem. Let $B = W_3^{(k+1)} + P^{(k)}$; then

$$R_2^{(k+1)} = CD^\top, \tag{19}$$

where $C$ and $D$ are obtained from the SVD $W_2^{(k+1)^\top} B = C\Sigma D^\top$.

**Updates for** $\Lambda$: Since $\Lambda = \mathrm{diag}(\lambda_1, \ldots, \lambda_r)$, we update each $\lambda_i$ in parallel by solving

$$\lambda_i^{(k+1)} = \operatorname*{arg\,min}_{\lambda_i} \frac{\gamma}{2} \left\| W_{21}^{(k+1)}(:,i) - \lambda_i W_{11}^{(k+1)}(:,i) + N_1^{(k)}(:,i) \right\|_F^2$$
$$+ \frac{\gamma}{2} \left\| W_{22}^{(k+1)}(:,i) - \frac{1}{\lambda_i} W_{12}^{(k+1)}(:,i) + N_2^{(k)}(:,i) \right\|_F^2.$$

Setting the derivative with respect to $\lambda_i$ to zero yields the quartic equation

$$\lambda_i^4 \| W_{11}^{(k+1)}(:,i) \|_2^2 - \lambda_i^3 \left( W_{21}^{(k+1)}(:,i) + N_1^{(k)}(:,i) \right)^\top W_{11}^{(k+1)}(:,i)$$
$$+ \lambda_i \left( W_{22}^{(k+1)}(:,i) + N_2^{(k)}(:,i) \right)^\top W_{12}^{(k+1)}(:,i) - \| W_{12}^{(k+1)}(:,i) \|_2^2 = 0,$$

which can be solved efficiently for $\lambda_i$.

## B. Optimal step size for ascent

Updating the nonnegative entries of $U$ in any given row does not affect the partial derivatives of the cost function for nonnegative entries of $U$ in other rows, and we can leverage this to update nonnegative entries in the rows of $U$ independently. The updates for all the nonnegative entries in the $i^{\text{th}}$ row are computed as

$$U_{ij} \leftarrow \left( U_{ij} - d_i^\star (\nabla_U f)_{ij} \right)_+, \quad \forall (i,j) \in I_+, \tag{20}$$

where $I_+$ denotes the set of nonnegative coordinates, $\nabla_U f = (UV^\top - X)V$, $\nabla_V f = (UV^\top - X)^\top U$, $d_i^\star, t_i^\star$ are the optimal step sizes, and the projection operator $(x)_+ = \max\{0, x\}$ ensures nonnegativity. To find the optimal step size, we introduce the error matrix at iteration $(k+1)$: $E^{(k+1)} = X - U^{(k+1)}V^{(k+1)^\top}$. Then, with the above definitions, we formulate the optimization problem as

$$\operatorname*{minimize}_{d_i} \quad \left\| X_{i,:} - \left( U_{i,:}^{(k)} - d_i \left[ M_{U_+} \circ \nabla_U f \right]_{i,:} \right) V^{(k)^\top} \right\|_F^2. \tag{21}$$

where $M_{U_+}$ is the binary mask over the nonnegative entries of $U$. The problem in (21) is an unconstrained quadratic optimization and has a global minimum, which can be computed by setting the gradient with respect to $d_i$ to zero:

$$\nabla_{d_i} L = \sum_{j=1}^n \left( 2E^{(k)}(i,j)a_j + 2a_j^2 d_i \right),$$

where $a_j = \{M_{U_+} \circ \nabla_U f\}(i,:) V^{(k)^\top}(:,j)$. Solving $\nabla_{d_i} L = 0$, we obtain

$$d_i^\star = \frac{-\sum_{j=1}^n E^{(k)}(i,j)a_j}{\sum_{j=1}^n a_j^2}.$$

The expression for $d_i^\star$ can be further simplified to

$$d_i^\star = \frac{\| \{M_{U_+} \circ \nabla_U f\}(i,:) \|^2}{\| \{M_{U_+} \circ \nabla_U f\}(i,:) V^{(k)^\top} \|^2}. \tag{22}$$

## C. Exterior NMC algorithm

The eNMF algorithm is extended to the case where the data matrix $X$ has missing entries. We compute a local minimum of the unconstrained problem

$$\underset{U,V}{\text{minimize}} \quad \hat{f}(U,V) = \tfrac{1}{2}\|M_E \circ (X - UV^\top)\|_F^2,$$

where $M_E$ is the binary mask over the unknown entries of $X$, using an alternating least squares algorithm (softImpute-ALS) proposed in (Hastie et al., 2015). Once we obtain a local minimum, we follow the same sequence (i.e., orthogonal transformation of $(U,V)$ closest to the positive orthant, followed by an exterior penalty method) to obtain a solution to the nonnegative matrix completion problem. The gradient computations in the exterior penalty method are updated accordingly:

$$\nabla_U \hat{f} = (M_E \circ (UV^\top - X))V, \quad \nabla_V \hat{f} = (M_E^\top \circ (UV^\top - X)^\top)U.$$

The pseudocode for eNMC is given below:

---

**Algorithm 5** eNMC

---

1: **Input**: $X, r, M_E, \epsilon,$ `max_iter`
2: $U_0, V_0 \leftarrow \text{softImpute}(X, r, M_E)$
3: $(U_0 R, V_0 R) \leftarrow \text{orthogonal}(U_0, V_0, R^0, Y^0, \rho)$
4: $(U, V) \leftarrow (U_0 R, V_0 R)$
5: **Repeat until convergence**:
6:     Update negative elements in $U$ using Eq. 6
7:     Compute mask $M_{U_+}$
8:     $U \leftarrow \text{PBCD}(X, U, V, \epsilon, \texttt{max\_iter}, M_{U_+}, M_E)$
9:     Update negative elements in $V$ using Eq. 7
10:    Compute mask $M_{V_+}$
11:    $V \leftarrow \text{PBCD}(X^\top, V, U, \epsilon, \texttt{max\_iter}, M_{V_+}, M_E^\top)$
12: **Output**: $U, V$

---

## D. Equal-time constrained Reconstruction error statistics

*Table 3.* **Equal-time constrained Relative reconstruction errors on synthetic datasets.**

| SNR | $r$ | eNMF | HALS | NMF-ADMM | Grad-Mult | ALS | AO-ADMM | A-HALS | NeNMF | FPGM | Vavasis |
|---|---|---|---|---|---|---|---|---|---|---|---|
| | 50 | **1.00** | 1.05 | 1.05 | 1.06 | 1.05 | 1.05 | 1.05 | 1.05 | 1.05 | 1.05 |
| | 100 | **1.00** | 1.05 | 1.06 | 1.07 | 1.05 | 1.05 | 1.05 | 1.05 | 1.05 | 1.05 |
| 20dB | 200 | **1.00** | 1.08 | 1.09 | 1.1 | 1.08 | 1.08 | 1.08 | 1.07 | 1.08 | 1.07 |
| | 400 | **1.00** | 1.12 | 1.14 | 1.15 | 1.12 | 1.11 | 1.11 | 1.1 | 1.11 | 1.13 |
| | 500 | **1.00** | 1.21 | 1.21 | 1.23 | 1.18 | 1.18 | 1.19 | 1.17 | 1.18 | 1.16 |
| | 50 | **1.00** | 1.05 | 1.06 | 1.07 | 1.05 | 1.05 | 1.05 | 1.05 | 1.05 | 1.06 |
| | 100 | **1.00** | 1.06 | 1.08 | 1.11 | 1.07 | 1.06 | 1.06 | 1.06 | 1.07 | 1.07 |
| 40dB | 200 | **1.00** | 1.09 | 1.09 | 1.14 | 1.07 | 1.07 | 1.08 | 1.06 | 1.08 | 1.08 |
| | 400 | **1.00** | 1.14 | 1.14 | 1.19 | 1.14 | 1.13 | 1.13 | 1.11 | 1.11 | 1.13 |
| | 500 | **1.00** | 1.25 | 1.22 | 1.27 | 1.21 | 1.21 | 1.22 | 1.19 | 1.2 | 1.18 |
| | 50 | **1.00** | 1.08 | 1.09 | 1.11 | 1.07 | 1.07 | 1.07 | 1.07 | 1.07 | 1.08 |
| | 100 | **1.00** | 1.09 | 1.1 | 1.12 | 1.08 | 1.08 | 1.08 | 1.07 | 1.08 | 1.1 |
| 60dB | 200 | **1.00** | 1.15 | 1.14 | 1.14 | 1.11 | 1.11 | 1.12 | 1.11 | 1.12 | 1.12 |
| | 400 | **1.00** | 1.22 | 1.22 | 1.23 | 1.2 | 1.19 | 1.2 | 1.19 | 1.21 | 1.22 |
| | 500 | **1.00** | 1.3 | 1.27 | 1.29 | 1.25 | 1.23 | 1.26 | 1.22 | 1.23 | 1.22 |
| | 50 | **1.00** | 1.1 | 1.11 | 1.12 | 1.09 | 1.1 | 1.1 | 1.09 | 1.09 | 1.1 |
| | 100 | **1.00** | 1.13 | 1.14 | 1.15 | 1.10 | 1.10 | 1.11 | 1.10 | 1.12 | 1.12 |
| 80dB | 200 | **1.00** | 1.21 | 1.24 | 1.27 | 1.18 | 1.19 | 1.2 | 1.17 | 1.2 | 1.2 |
| | 400 | **1.00** | 1.27 | 1.29 | 1.31 | 1.23 | 1.23 | 1.24 | 1.22 | 1.23 | 1.24 |
| | 500 | **1.00** | 1.33 | 1.35 | 1.38 | 1.27 | 1.25 | 1.29 | 1.25 | 1.28 | 1.26 |
| | 50 | **1.00** | 1.11 | 1.13 | 1.14 | 1.10 | 1.09 | 1.10 | 1.10 | 1.11 | 1.12 |
| | 100 | **1.00** | 1.13 | 1.16 | 1.18 | 1.1 | 1.1 | 1.11 | 1.09 | 1.1 | 1.14 |
| 100dB | 200 | **1.00** | 1.2 | 1.22 | 1.25 | 1.15 | 1.14 | 1.16 | 1.14 | 1.17 | 1.19 |
| | 400 | **1.00** | 1.27 | 1.29 | 1.29 | 1.21 | 1.2 | 1.23 | 1.18 | 1.2 | 1.27 |
| | 500 | **1.00** | 1.35 | 1.38 | 1.4 | 1.29 | 1.28 | 1.3 | 1.27 | 1.3 | 1.29 |

**Note.** Identical wall-clock budgets for all methods.

*Table 4.* **Equal-time constrained NMF reconstruction errors on real datasets.**

| Dataset | $r$ | eNMF | HALS | NMF-ADMM | Grad-Mult | ALS | AO-ADMM | A-HALS | NeNMF | FPGM | Vavasis |
|---|---|---|---|---|---|---|---|---|---|---|---|
| | 5 | **18653.23** | 18708.23 | 18708.23 | 18969.8 | 18708.23 | 18708.23 | 18708.23 | 18708.23 | 18708.23 | 18708.23 |
| | 10 | **12400.48** | 12570.38 | 14798.48 | 13791.78 | 12550.75 | 12580.30 | 12565.29 | 12521.70 | 12541.96 | 13761.22 |
| Face | 15 | **9579.31** | 9805.9 | 12220.89 | 11312.24 | 9926.48 | 9823.40 | 9802.11 | 9746.81 | 9922.37 | 11340.66 |
| | 20 | **7234.27** | 7939.04 | 22587.43 | 10171.16 | 7941.21 | 7960.50 | 7930.16 | 7899.33 | 7936.88 | 8187.6 |
| | 25 | **5953.34** | 6355.44 | 20205.44 | 8368.67 | 6430 | 6372.65 | 6333.67 | 6218.87 | 6428.77 | 7170 |
| | 10 | **13.59** | 14.39 | 14.48 | 14.41 | 14.39 | 14.39 | 14.38 | 14.39 | 14.39 | 14.41 |
| | 20 | **12.72** | 13.46 | 13.95 | 13.61 | 13.60 | 13.59 | 13.46 | 13.52 | 13.57 | 13.54 |
| Verb | 40 | **11.63** | 12.32 | 13.99 | 12.38 | 12.32 | 12.34 | 12.29 | 12.33 | 12.31 | 12.37 |
| | 80 | **9.81** | 10.53 | 16.05 | 10.67 | 10.48 | 10.53 | 10.51 | 10.52 | 10.48 | 11.25 |
| | 100 | **8.97** | 9.74 | 17.17 | 9.94 | 9.7 | 9.77 | 9.69 | 9.7 | 9.7 | 9.83 |
| | 10 | **9365.11** | 9915.16 | 12190.72 | 12637.65 | 9912.53 | 9810.46 | 9915.16 | 9796.35 | 9911.29 | 10643.59 |
| | 20 | **9140.42** | 9612.82 | 11496.51 | 12147.03 | 9485.06 | 9486.61 | 9610.12 | 9485.66 | 9485.06 | 10140.9 |
| Audio | 40 | **8936.93** | 9290.37 | 11460.15 | 11986.23 | 9212.37 | 9082.16 | 9282.67 | 9066.1 | 9201.87 | 9956.82 |
| | 80 | **7942.84** | 8566.2 | 10708.93 | 11584.87 | 9053.49 | 8402.73 | 8565.9 | 8376.42 | 8969.14 | 9876.69 |
| | 100 | **7782.36** | 8497.58 | 10585.91 | 11032.56 | 8938.46 | 8185.34 | 8497.58 | 8123.34 | 8937.2 | 9155.27 |

**Note.** Identical wall-clock budgets for all methods.

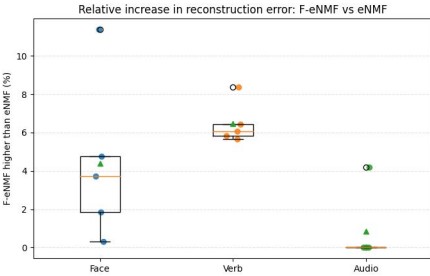

*Figure 5.* **Feasible eNMF (F-eNMF) (eNMF at feasibility before descent to local minima) hits very close to the local minima**: We computed the relative increase in the reconstruction error of F-eNMF and eNMF. Across three real datasets, F-eNMF have up to 12% higher reconstruction error than eNMF.

# E. Downstream tasks

### E.1. Face Recognition Experiment

To evaluate the effectiveness of the learned basis from our NMF algorithm on a downstream task, we conducted a **face recognition experiment** using the face dataset. This experiment demonstrates that the representations obtained from our method not only reduce reconstruction error but also capture discriminative features useful for identifying subjects under varying illumination conditions.

#### E.1.1. DATASET PREPARATION

The face dataset contains grayscale images of multiple subjects under different illumination and pose conditions. Each image was vectorized into a column vector of pixel intensities, resulting in a data matrix $X \in \mathbb{R}^{32256 \times 64}$. The corresponding subject ID of each image was used as the label for recognition.

Prior to the experiment, all images were normalized to maintain nonnegativity, and a train-test split was performed. We used a **stratified split** to ensure that each subject's images were proportionally represented in both training and test sets, with 70% of images for training and 30% for testing. Multiple splits were used to report mean and standard deviation of recognition performance.

#### E.1.2. FEATURE EXTRACTION VIA NMF PROJECTION

For each method (eNMF and other NMF algorithms), we obtained a basis matrix $W \in \mathbb{R}^{32256 \times r}$ from the face dataset. To extract features for the face dataset, each test and training image $x \in \mathbb{R}^{32256}$ was **projected onto the learned basis** by solving the following nonnegative least squares (NNLS) problem:

$$h = \arg\min_{h \geq 0} \|x - Uh\|_2^2,$$

where $h \in \mathbb{R}^r$ is the coefficient vector representing the image in the basis space. Collecting all coefficient vectors formed the feature matrix $H \in \mathbb{R}^{r \times 64}$ for training and testing.

This projection ensures that the features respect the nonnegativity constraints of NMF and capture the parts-based representations learned from the original dataset.

#### E.1.3. CLASSIFIER TRAINING

Once the feature vectors were extracted, we trained a **support vector machine (SVM)** with an RBF kernel to classify images by subject ID. Training was performed on the projected training set using the corresponding labels. Hyperparameters for the SVM were tuned via cross-validation on the training folds.

#### E.1.4. EVALUATION

The trained classifier was applied to the projected test set to predict subject IDs. Recognition performance was measured as **classification accuracy** (percentage of correctly classified images). To ensure robustness, the experiment was repeated across multiple random train-test splits, and the mean and standard deviation of the accuracy were reported.

#### E.1.5. RESULTS

This recognition experiment demonstrates that the features obtained from eNMF algorithm are not only effective for low-rank reconstruction but also **highly discriminative for downstream tasks**. Compared to baseline NMF methods, our algorithm consistently achieved higher recognition accuracy on the face dataset, validating the quality and transferability of the learned representations (see Table 5 for results).

### E.2. AudioMNIST Experiment

To evaluate the effectiveness of the learned basis from our eNMF algorithm on a downstream task, we conducted an **audio digit recognition experiment** using the AudioMNIST dataset (Becker et al., 2023). This experiment demonstrates that the representations obtained from our method not only reduce reconstruction error but also capture discriminative features

| Rank ($r$) | eNMF | HALS | NMF-ADMM | Grad-Mult | ALS | AO-ADMM | A-HALS | NeNMF | FPGM | Vavasis |
|---|---|---|---|---|---|---|---|---|---|---|
| 5 | **81.6% ± 3.7%** | 61.2% ± 4.2% | 53.0% ± 4.7% | 57.1% ± 4.0% | 62.8% ± 3.9% | 75.1% ± 4.1% | 71.8% ± 4.0% | 68.5% ± 4.6% | 63.6% ± 4.4% | 66.1% ± 4.3% |
| 10 | **89.3% ± 2.1%** | 71.4% ± 2.6% | 62.5% ± 3.1% | 53.6% ± 3.5% | 65.2% ± 2.5% | 83.0% ± 2.4% | 81.3% ± 2.3% | 78.6% ± 2.7% | 72.3% ± 3.0% | 74.1% ± 2.8% |
| 15 | **93.8% ± 1.6%** | 76.9% ± 1.9% | 59.1% ± 2.0% | 68.5% ± 2.1% | 70.3% ± 1.8% | 81.6% ± 2.0% | 79.7% ± 1.9% | 83.5% ± 1.7% | 71.3% ± 2.2% | 80.7% ± 1.8% |
| 20 | **95.0% ± 1.4%** | 70.3% ± 1.8% | 68.4% ± 1.7% | 62.7% ± 1.9% | 74.1% ± 1.6% | 89.3% ± 1.6% | 85.5% ± 1.5% | 78.9% ± 1.8% | 77.9% ± 1.7% | 76.0% ± 1.8% |
| 25 | **95.6% ± 1.2%** | 74.6% ± 1.5% | 64.1% ± 1.6% | 56.4% ± 1.8% | 69.8% ± 1.7% | 87.0% ± 1.4% | 88.0% ± 1.3% | 82.2% ± 1.5% | 73.6% ± 1.6% | 80.3% ± 1.5% |

*Table 5.* Equal-time constrained comparison of NMF algorithms across different ranks ($r$) on the Face dataset. Values show recognition classification accuracy. Best results per row are bolded.

*Table 6.* Equal-time constrained digit recognition accuracy (%) on AudioMNIST using NMF features. Higher is better.

| $r$ | eNMF | HALS | NMF-ADMM | Grad-Mult | ALS | AO-ADMM | A-HALS | NeNMF | FPGM | Vavasis |
|---|---|---|---|---|---|---|---|---|---|---|
| 10 | **85.6 ± 2.3** | 62.5 ± 3.4 | 55.6 ± 3.0 | 47.1 ± 4.6 | 65.1 ± 2.8 | 75.3 ± 2.6 | 72.8 ± 2.9 | 70.2 ± 3.1 | 63.3 ± 3.5 | 66.8 ± 3.0 |
| 20 | **89.2 ± 1.9** | 69.6 ± 2.6 | 63.3 ± 2.4 | 55.3 ± 3.9 | 65.1 ± 2.7 | 80.3 ± 2.2 | 74.0 ± 2.4 | 76.7 ± 2.1 | 70.5 ± 2.8 | 68.7 ± 2.7 |
| 40 | **90.1 ± 1.6** | 63.1 ± 2.2 | 54.1 ± 2.0 | 53.2 ± 2.6 | 69.4 ± 1.9 | 76.6 ± 2.0 | 80.2 ± 1.8 | 72.1 ± 2.1 | 64.0 ± 2.3 | 74.8 ± 1.9 |
| 80 | **94.7 ± 1.1** | 72.0 ± 1.9 | 64.4 ± 2.1 | 48.3 ± 2.4 | 65.3 ± 2.0 | 86.2 ± 1.5 | 81.4 ± 1.8 | 83.3 ± 1.6 | 75.8 ± 1.7 | 71.0 ± 2.0 |
| 100 | **96.5 ± 0.9** | 76.2 ± 1.7 | 69.5 ± 1.9 | 61.8 ± 2.0 | 71.4 ± 1.8 | 81.1 ± 1.6 | 79.1 ± 1.5 | 84.0 ± 1.4 | 70.4 ± 1.8 | 75.3 ± 1.7 |

useful for recognizing spoken digits.

### E.2.1. DATASET PREPARATION

The AudioMNIST dataset contains 30,000 audio recordings of digits 0–9 from 60 speakers. Each speaker pronounced each digit 50 times, resulting in short audio clips of approximately 1 second. The raw audio was converted to log-magnitude spectrograms using a short-time Fourier transform (STFT) with a 25 ms window and 10 ms hop size. These spectrograms form the input data matrix for factorization.

The dataset was split into 80% training and 20% testing samples, and 5 random splits were used to report mean and standard deviation of performance.

### E.2.2. FEATURE EXTRACTION VIA NMF PROJECTION

For each method (eNMF and baseline NMF algorithms), the spectrograms were factorized into a basis matrix $U$ and coefficient matrix $V^\top$ with latent ranks $r \in \{10, 20, 40, 80, 100\}$. The coefficient vectors $V^\top$ were used as feature representations for downstream classification.

### E.2.3. EVALUATION TASK

We evaluated the algorithms on **Downstream digit recognition**, using a logistic regression classifier trained on the NMF features to predict the digit labels.

### E.2.4. RESULTS

**Downstream Digit Recognition.** Table 6 reports classification accuracy on the logistic regression task. eNMF consistently outperforms all baselines, demonstrating that its learned features are more discriminative for audio digit recognition.

Overall, this experiment demonstrates that eNMF provides both improved reconstruction quality and highly discriminative features for downstream classification tasks on audio data.

### E.3. Post-Rotation Ablations for AudioMNIST

### E.4. Recommendation experiment

We evaluate the proposed eNMC algorithm on the Movielens 1M dataset (Harper & Konstan, 2016), which consists of 1,000,209 ratings from 6,040 users on 3,706 items. The resulting rating matrix $X \in \mathbb{R}^{6040 \times 3706}$ is highly sparse, with approximately 95.5% missing entries. This dataset is widely used as a benchmark for collaborative filtering and recommendation tasks.

*Table 7.* **Equal-error constrained digit-recognition accuracy (%) on AudioMNIST using NMF features from different post-rotation strategies.** Starting from the same rotated SVD solution, we compare five post-rotation variants under the equal-error protocol: (i) our proposed feasibility-attainment stage followed by HALS descent, (ii) direct projection followed by HALS descent, (iii) direct projection followed by Grad-Mult descent, (iv) direct projection followed by standard gradient descent, and (v) our feasibility-attainment stage followed by standard gradient descent. **Key finding: under the equal-error protocol, the downstream quality of the solutions returned by these different post-rotation strategies is equivalent**. Please refer to Appendix E.2 for details on AudioMNIST digit-recognition experiment setup. Values are reported as mean $\pm$ standard deviation in percent; higher is better.

| $r$ | Feasibility + HALS Descent (Ours) | Projection + HALS Descent | Projection + Grad-Mult Descent | Projection + Gradient Descent | Feasibility + Gradient Descent |
|---|---|---|---|---|---|
| 10 | **85.6 $\pm$ 2.3** | 85.5 $\pm$ 2.2 | 85.4 $\pm$ 2.3 | 85.2 $\pm$ 2.4 | 85.4 $\pm$ 2.2 |
| 20 | **89.2 $\pm$ 1.9** | 89.2 $\pm$ 2 | 89.1 $\pm$ 1.6 | 88.9 $\pm$ 2.1 | 89 $\pm$ 1.8 |
| 40 | **90.1 $\pm$ 1.6** | 90.1 $\pm$ 1.6 | 90.1 $\pm$ 1.6 | 90.1 $\pm$ 1.6 | 90.1 $\pm$ 1.6 |
| 80 | **94.7 $\pm$ 1.1** | 94.7 $\pm$ 1.1 | 94.7 $\pm$ 1.1 | 94.7 $\pm$ 1.1 | 94.7 $\pm$ 1.1 |
| 100 | **96.5 $\pm$ 0.9** | 96.5 $\pm$ 0.9 | 96.5 $\pm$ 0.9 | 96.5 $\pm$ 0.9 | 96.5 $\pm$ 0.9 |

*Table 8.* **Equal-time constrained digit-recognition accuracy (%) on AudioMNIST using NMF features from different post-rotation strategies.** Starting from the same rotated SVD solution, we compare five post-rotation variants under a common wall-clock budget: (i) our proposed feasibility-attainment stage followed by HALS descent, (ii) direct projection followed by HALS descent, (iii) direct projection followed by Grad-Mult descent, (iv) direct projection followed by standard gradient descent, and (v) our feasibility-attainment stage followed by gradient descent. **Key finding: under the same time budget, our feasibility + HALS strategy produces substantially better downstream features at lower and moderate ranks, while all strategies become identical at higher ranks where the rotated SVD solution is already optimal.** The gains at $r = 10$ and 20 show that the proposed ascent/PBCD stage preserves and refines the rotated low-rank structure much more effectively than forceful projection when computation is limited. For AudioMNIST at $r = 40, 80, 100$, however, the rotated SVD solution already attains the unconstrained global minimum, so all post-rotation strategies start from the same optimal point and therefore yield identical downstream classification accuracy. Please refer to Appendix E.2 for details on AudioMNIST digit-recognition experiment setup. Values are reported as mean $\pm$ standard deviation in percent; higher is better.

| $r$ | Feasibility + HALS Descent (Ours) | Projection + HALS Descent | Projection + Grad-Mult Descent | Projection + Gradient Descent | Feasibility + Gradient Descent |
|---|---|---|---|---|---|
| 10 | **85.6 $\pm$ 2.3** | 75.4 $\pm$ 1.9 | 74.2 $\pm$ 2.1 | 72.9 $\pm$ 2.4 | 75.1 $\pm$ 1.8 |
| 20 | **89.2 $\pm$ 1.9** | 80.4 $\pm$ 2.2 | 78.9 $\pm$ 2.1 | 77.4 $\pm$ 1.9 | 80.3 $\pm$ 2 |
| 40 | **90.1 $\pm$ 1.6** | 90.1 $\pm$ 1.6 | 90.1 $\pm$ 1.6 | 90.1 $\pm$ 1.6 | 90.1 $\pm$ 1.6 |
| 80 | **94.7 $\pm$ 1.1** | 94.7 $\pm$ 1.1 | 94.7 $\pm$ 1.1 | 94.7 $\pm$ 1.1 | 94.7 $\pm$ 1.1 |
| 100 | **96.5 $\pm$ 0.9** | 96.5 $\pm$ 0.9 | 96.5 $\pm$ 0.9 | 96.5 $\pm$ 0.9 | 96.5 $\pm$ 0.9 |

### E.4.1. BASELINES

We benchmark eNMC against three widely used nonnegative matrix completion algorithms:

- **Mult**: multiplicative updates (Lin & Boutros, 2018),

- **SCD**: sequential coordinate descent (Lin & Boutros, 2018),

- **ADM**: alternating direction method (Xu et al., 2012).

### E.4.2. EVALUATION METRICS

We consider reconstruction error along with two types of downstream tasks:

1. **Rating prediction:** Root Mean Squared Error (RMSE) is computed on the held-out test ratings.

2. **Top-$K$ recommendation:** For each user, unobserved entries are scored and ranked. We report Recall@K, and NDCG@K, for $K \in \{10\}$. Training items are excluded when ranking.

### E.4.3. RECONSTRUCTION ERROR

### E.4.4. EXPERIMENTAL SETUP FOR DOWNSTREAM TASKS

For each user, we randomly split the observed ratings into $80\%$ training, $10\%$ validation, and $10\%$ test sets. To simulate realistic recommendation settings, we also adopt a leave-one-out protocol for the ranking evaluation: for each user, one observed rating is held out for testing and one for validation, with the rest used for training. We evaluate matrix factorizations with latent dimensions $r \in \{5, 10, 15, 25\}$. Hyperparameters (e.g., regularization strength) are tuned on the validation set. Each result is reported as mean $\pm$ standard deviation over five random initialization seeds.

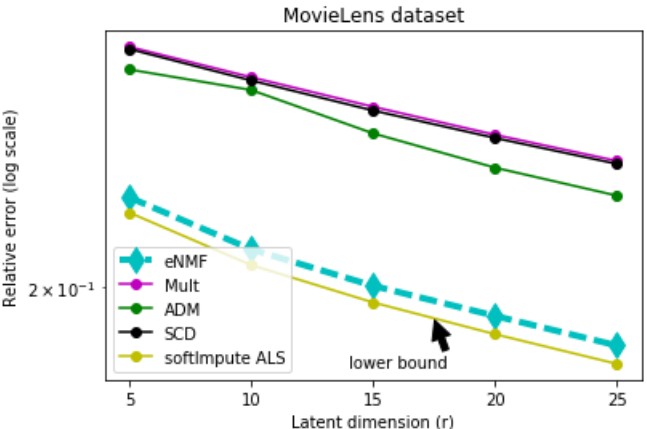

*Figure 6.* **Equal-time constrained NMC errors**: Relative reconstruction error (RE) $\frac{\|P_\Omega(X-UV^\top)\|_F}{\|P_\Omega X\|_F}$ is plotted as a function of the number of latent dimensions ($r$). *eNMC achieves the lowest reconstruction error for all latent dimensions.*

### E.4.5. STATISTICAL TESTING

For rating prediction, significance is assessed using paired $t$-tests across random seeds. For ranking metrics, we conduct per-user Wilcoxon signed-rank tests comparing eNMC against each baseline. Improvements with $p < 0.05$, $p < 0.01$, and $p < 0.001$ are denoted by *, **, and ***, respectively.

### E.4.6. RESULTS

| Rank $r$ | eNMC | Mult | SCD | ADM |
|---|---|---|---|---|
| 5 | **0.825 $\pm$ 0.004**\*\*\* | 0.939 $\pm$ 0.006 | 0.923 $\pm$ 0.005 | 0.918 $\pm$ 0.007 |
| 10 | **0.817 $\pm$ 0.003**\*\*\* | 0.905 $\pm$ 0.005 | 0.9 $\pm$ 0.004 | 0.896 $\pm$ 0.006 |
| 15 | **0.785 $\pm$ 0.002**\*\*\* | 0.879 $\pm$ 0.004 | 0.844 $\pm$ 0.003 | 0.852 $\pm$ 0.005 |
| 25 | **0.758 $\pm$ 0.003**\*\*\* | 0.869 $\pm$ 0.004 | 0.862 $\pm$ 0.003 | 0.858 $\pm$ 0.004 |

*Table 9.* Equal-time constrained Rating prediction (RMSE) on Movielens 1M. Mean $\pm$ std over 5 random seeds. Best result in bold; significance markers indicate eNMC improvements over all baselines. Lower the better

| Rank $r$ | eNMC (NDCG@10) | Best baseline | eNMC (Recall@10) | Best baseline |
|---|---|---|---|---|
| 5 | **0.419 $\pm$ 0.004**\*\* | 0.289 (ADM) | **0.694 $\pm$ 0.005**\*\*\* | 0.567 (ADM) |
| 10 | **0.496 $\pm$ 0.004**\*\*\* | 0.318 (ADM) | **0.748 $\pm$ 0.005**\*\*\* | 0.588 (ADM) |
| 15 | **0.573 $\pm$ 0.005**\*\*\* | 0.335 (SCD) | **0.778 $\pm$ 0.004**\*\*\* | 0.6 (SCD) |
| 25 | **0.687 $\pm$ 0.006**\*\*\* | 0.344 (ADM) | **0.832 $\pm$ 0.004**\*\*\* | 0.627 (ADM) |

*Table 10.* Equal-time constrained Top-$K$ recommendation on Movielens 1M (NDCG@10 and Recall@10). Mean $\pm$ std over 5 seeds. Best baseline is shown in parentheses. Higher the better.

In both rating prediction (Table 9 and top-$K$ recommendation tasks (Table 10, eNMC consistently outperforms the baselines across all latent dimensions, with statistically significant improvements in most configurations.

## F. Generalized orthogonal transformation

Given eNMF and another algorithm that reaches the same reconstruction error, i.e., $U_{\text{alg}}V_{\text{alg}}^\top = U_{\text{eNMF}}V_{\text{eNMF}}^\top$, we seek $R_1$, $R_2$, and $\Lambda$ such that the matrix factors from different algorithms are related by the generalized orthogonal transformation

$$U_{\text{alg}} = U_{\text{eNMF}} R_1 \Lambda R_2^\top,$$
$$V_{\text{alg}} = V_{\text{eNMF}} R_1 \Lambda^{-1} R_2^\top,$$

| r | SNR | eNMF vs AO-ADMM: $(\delta_u, \delta_v)$ |
|---|---|---|
| 100 | 20$dB$ | $(10^{-10}, 10^{-10})$ |
| | 40$dB$ | $(10^{-20}, 10^{-20})$ |
| | 60$dB$ | $(10^{-20}, 10^{-20})$ |
| | 80$dB$ | $(10^{-25}, 10^{-25})$ |
| | 100$dB$ | $(10^{-20}, 10^{-20})$ |

*Table 11.* **Equivalence of Factor Matrices (scaled and rotated versions of each other) for Synthetic datasets**: We used the above algorithm to compute the generalized orthogonal parameters, $R_1$, $R_2$ and $\Lambda$. We verified that $R_1 \approx I$ by computing $\|R_1 - I\|_F$ for all 5 SNR levels ($\|R_1 - I\|_F \leq 10^{-6}$). We then computed the residual errors $(\delta_u, \delta_v)$, where $\delta_u = \|U_{alg} - U_{eNMF}\Lambda R_2^T\|_F$ and $\delta_v = \|V_{alg} - V_{eNMF}\Lambda^{-1}R_2^T\|_F$, and found that $(\delta_u \approx 0, \delta_v \approx 0)$. Thus, the NMF factors obtained from the AO-ADMM algorithm are scaled and rotated versions of those obtained by the eNMF algorithm. Thus, *the feature vectors retain the same pairwise cosine distances, and are equivalent from a clustering perspective.*

and such a transformation always exists (Vavasis, 2009; Stewart, 2001). Following a similar approach as in Section A.1, we can find $R_1$, $R_2$, and $\Lambda$ by computing the SVD of $Q\Gamma^{-1}P^\top U_{\text{alg}}$, where

$$\text{SVD}(U_{\text{eNMF}}) = P\Gamma Q^\top.$$

We applied this algorithm to determine whether for the case of synthetic data –where eNMF achives globally optimal objective value, and AO-ADMM converges to a value very close to this optimum - the factor matrices computed by the two algorithms are equivalent. As shown in Table 11, the factor matrices are indeed scaled and rotated versions of each other, with very small residual errors. The AO-ADMM algorithm only asymptotically approaches globally optimum reconstruction error, accounting for the observed residual errors. Thus the non-negative factors generated by both eNMF and asymptotically by AO-ADMM (which takes longer time to converge than eNMF) preserve the same pair-wise feature distance metrics (in this case cosine similarity) as obtained by performing unconstrained optimization using SVD.

## G. KKT Optimality Conditions

**KKT optimality conditions.** For the objective $f(U, V)$ (e.g., $f(U, V) = \frac{1}{2}\|X - UV^\top\|_F^2$), the KKT conditions for the nonnegativity constraints $U \geq 0$, $V \geq 0$ are:

$$
\begin{aligned}
U \geq 0, \quad \nabla_U f(U, V) \geq 0, \quad \big(\nabla_U f(U, V)\big) \odot U = 0, \\
V \geq 0, \quad \nabla_V f(U, V) \geq 0, \quad \big(\nabla_V f(U, V)\big) \odot V = 0.
\end{aligned}
\tag{23}
$$

*Table 12.* **KKT verification for real and synthetic experiments:** We tabulate the tuple $(\delta_W, \sigma_W)$, where $\delta_W = \max(\nabla W \odot W)$ and $\sigma_W = \sum_{i,j}(\nabla W \odot W)_{ij}$. As before, $W$ is the stacked NMF factors, i.e., $W = \begin{bmatrix} U & V \end{bmatrix}^\top$. eNMF satisfies the KKT conditions $(\delta_W \approx 0, \sigma_W \approx 0)$ for all settings.

| | | | | | | **(I) Real datasets (equal-error constrained experiments)** | | | | | |
|---|---|---|---|---|---|---|---|---|---|---|---|
| Dataset | r | eNMF | HALS | NMF-ADMM | Grad-Mult | ALS | AO-ADMM | A-HALS | NeNMF | FPGM | Vavasis |
| Face | 5 | (0,0) | (0,0) | (0,0) | (0,0) | (0,0) | (0,0) | (0,0) | (0,0) | (0,0) | (0,0) |
| | 10 | (0,0) | (0,0) | (1.7,5.3) | (0.9,1.4) | (0,0) | (0,0) | (0,0) | (0,0) | (0,0) | (0,0) |
| | 15 | (0,0) | (0.06,0.1) | (0,0) | (0,0) | (0.05,0.11) | (0,0) | (0,0) | (0,0) | (0.02,0.06) | (0.03,0.06) |
| | 20 | (0,0) | (0,0) | (0,0) | (0,0) | (0,0) | (0,0) | (0,0) | (0,0) | (0,0) | (0,0) |
| | 25 | (0,0) | (0,0) | (0,0) | (0,0) | (0,0) | (0,0) | (0,0) | (0,0) | (0,0) | (0,0) |
| Verb | 10 | (0,0) | (0.008,0.009) | (0.06,0.13) | (0.03,0.08) | (0.003,0.005) | (0.004,0.005) | (0.008,0.009) | (0.005,0.005) | (0.006,0.009) | (0.007,0.009) |
| | 20 | (0,0) | (0.009,0.009) | (0.08,0.15) | (0.07,0.1) | (0.004,0.005) | (0.004,0.005) | (0.008,0.009) | (0.005,0.005) | (0.007,0.009) | (0.007,0.008) |
| | 40 | (0,0) | (0.0002,0.0003) | (0.006,0.009) | (0.007,0.009) | (0.001,0.003) | (0,0) | (0.0003,0.0003) | (0,0) | (0.002,0.003) | (0.001,0.003) |
| | 80 | (0,0) | (0,0) | (0.01,0.03) | (0.01,0.03) | (0.008,0.01) | (0,0) | (0,0) | (0,0) | (0.009,0.01) | (0.009,0.013) |
| | 100 | (0,0) | (0,0) | (0.008,0.009) | (0.007,0.009) | (0.004,0.006) | (0,0) | (0,0) | (0,0) | (0.001,0.001) | (0.007,0.01) |
| Audio | 10 | (0,0) | (0.51,0.66) | (0.4,0.55) | (0.34,0.7) | (0.38,0.57) | (0.2,0.36) | (0.33,0.68) | (0.14,0.29) | (0.4,0.6) | (0.35,0.74) |
| | 20 | (0,0) | (0.1,0.21) | (0.12,0.27) | (0.12,0.28) | (0.08,0.12) | (0.08,0.1) | (0.1,0.2) | (0.13,0.22) | (0.12,0.19) | (0.14,0.25) |
| | 40 | (0,0) | (0,0) | (0.1,0.23) | (0.13,0.3) | (0,0) | (0,0) | (0,0) | (0,0) | (0,0) | (0,0) |
| | 80 | (0,0) | (0,0) | (0.02,0.07) | (0.02,0.05) | (0,0) | (0,0) | (0,0) | (0,0) | (0,0) | (0,0) |
| | 100 | (0,0) | (0,0) | (0,0) | (0,0) | (0,0) | (0,0) | (0,0) | (0,0) | (0,0) | (0,0) |

| | | | | | | **(II) Synthetic datasets (equal-error constrained experiments, SNR sweep)** | | | | | |
|---|---|---|---|---|---|---|---|---|---|---|---|
| SNR Level | r | eNMF | HALS | NMF-ADMM | Grad-Mult | ALS | AO-ADMM | A-HALS | NeNMF | FPGM | Vavasis |
| 20dB | 50 | (0,0) | (0.006,0.01) | (0.007,0.018) | (0.01,0.023) | (0.006,0.015) | (0,0) | (0,0) | (0.008,0.009) | (0.007,0.008) | (0.007,0.009) |
| | 100 | (0,0) | (0.005,0.014) | (0.006,0.021) | (0.014,0.029) | (0.004,0.017) | (0,0) | (0,0) | (0.002,0.011) | (0.001,0.009) | (0.001,0.01) |
| | 200 | (0,0) | (0.003,0.015) | (0.009,0.019) | (0.01,0.027) | (0.008,0.018) | (0,0) | (0.001,0.0012) | (0.003,0.012) | (0.002,0.011) | (0.004,0.015) |
| | 400 | (0,0) | (0.005,0.017) | (0.011,0.024) | (0.02,0.032) | (0.01,0.02) | (0,0) | (0.0015,0.0019) | (0.004,0.013) | (0.006,0.014) | (0.007,0.016) |
| | 500 | (0,0) | (0.009,0.02) | (0.01,0.025) | (0.013,0.03) | (0.009,0.022) | (0,0) | (0.0016,0.002) | (0.006,0.015) | (0.005,0.014) | (0.008,0.017) |
| 40dB | 50 | (0,0) | (0,0) | (0,0) | (0.14,0.32) | (0,0) | (0,0) | (0,0) | (0,0) | (0,0) | (0,0) |
| | 100 | (0,0) | (0,0) | (0,0) | (0.13,0.37) | (0,0) | (0,0) | (0,0) | (0,0) | (0,0) | (0,0) |
| | 200 | (0,0) | (0,0) | (0.2,0.44) | (0.18,0.51) | (0.09,0.22) | (0,0) | (0,0) | (0,0) | (0,0) | (0,0) |
| | 400 | (0,0) | (0,0) | (0.22,0.36) | (0.27,0.64) | (0.014,0.3) | (0,0) | (0,0) | (0,0) | (0,0) | (0,0) |
| | 500 | (0,0) | (0,0) | (0.25,0.47) | (0.3,0.76) | (0.32,0.58) | (0,0) | (0,0) | (0,0) | (0,0) | (0,0) |
| 60dB | 50 | (0,0) | (0,0) | (0.006,0.009) | (0.005,0.01) | (0.004,0.007) | (0,0) | (0,0) | (0,0) | (0,0) | (0,0) |
| | 100 | (0,0) | (0,0) | (0.003,0.009) | (0.008,0.014) | (0.002,0.007) | (0,0) | (0,0) | (0,0) | (0,0) | (0,0) |
| | 200 | (0,0) | (0,0) | (0.001,0.008) | (0.009,0.02) | (0.002,0.005) | (0,0) | (0,0) | (0,0) | (0,0) | (0,0) |
| | 400 | (0,0) | (0.001,0.001) | (0.004,0.01) | (0.011,0.02) | (0.006,0.011) | (0,0) | (0,0) | (0.001,0.002) | (0.002,0.002) | (0.001,0.001) |
| | 500 | (0,0) | (0.0009,0.002) | (0.003,0.014) | (0.009,0.016) | (0.006,0.01) | (0,0) | (0,0) | (0.0014,0.002) | (0.0011,0.0017) | (0.001,0.0018) |
| 80dB | 50 | (0,0) | (0,0) | (0,0) | (0.007,0.008) | (0,0) | (0,0) | (0,0) | (0,0) | (0,0) | (0,0) |
| | 100 | (0,0) | (0,0) | (0,0) | (0.1,0.2) | (0,0) | (0,0) | (0,0) | (0,0) | (0,0) | (0,0) |
| | 200 | (0,0) | (0,0) | (0.008,0.01) | (0.12,0.19) | (0,0) | (0,0) | (0,0) | (0,0) | (0,0) | (0,0) |
| | 400 | (0,0) | (0,0) | (0.15,0.26) | (0.16,0.33) | (0.05,0.09) | (0,0) | (0,0) | (0,0) | (0,0) | (0,0) |
| | 500 | (0,0) | (0,0) | (0.13,0.28) | (0.17,0.43) | (0.11,0.2) | (0,0) | (0,0) | (0,0) | (0,0) | (0,0) |
| 100dB | 50 | (0,0) | (0,0) | (0,0) | (0,0) | (0,0) | (0,0) | (0,0) | (0,0) | (0,0) | (0,0) |
| | 100 | (0,0) | (0,0) | (0,0) | (0,0) | (0,0) | (0,0) | (0,0) | (0,0) | (0,0) | (0,0) |
| | 200 | (0,0) | (0,0) | (0,0) | (0.01,0.09) | (0,0) | (0,0) | (0,0) | (0,0) | (0,0) | (0,0) |
| | 400 | (0,0) | (0,0) | (0.009,0.07) | (0.02,0.13) | (0.008,0.009) | (0,0) | (0,0) | (0,0) | (0,0) | (0,0) |
| | 500 | (0,0) | (0,0) | (0.03,0.1) | (0.05,0.15) | (0.01,0.09) | (0,0) | (0,0) | (0,0) | (0,0) | (0,0) |

Table 12 reports KKT verification for both our real-data equal-error experiments (Block I) and the synthetic SNR sweep (Block II) by summarizing complementary-slackness residuals $(\delta_W, \sigma_W)$, where $W = [U\,V]^\top$, $\delta_W = \max(\nabla W \odot W)$, and $\sigma_W = \sum_{i,j}(\nabla W \odot W)_{ij}$. Across all datasets/SNRs and all tested latent dimensions, eNMF attains $(0,0)$, indicating that it consistently reaches a KKT-consistent stationary point under the equal-error constraint. In contrast, competing methods satisfy the KKT conditions only sporadically: on real datasets, several baselines achieve $(0,0)$ for easier regimes (e.g., Face at $r \in \{5, 20, 25\}$ and Audio at larger ranks), but exhibit nonzero residuals at intermediate ranks (e.g., Face $r = 10$ for NMF-ADMM and Grad-Mult, and Face $r = 15$ for HALS/ALS/FPGM/Vavasis). Importantly, **even when nonzero**, the reported residuals are typically **small** (often on the order of $10^{-3}$–$10^{-1}$ in Block II), suggesting that many baselines are **not fully stationary within the budget but are trending toward stationarity** (i.e., approaching KKT-consistent points). On synthetic data, most methods are KKT-consistent in high-SNR settings (notably 100dB at $r \leq 100$), whereas at lower SNRs the residuals become more pronounced—especially for gradient-based updates (Grad-Mult) and, in some configurations, NMF-ADMM/ALS—highlighting the increased difficulty of meeting stationarity in noisier regimes. Overall, these results support that eNMF's reconstruction gains are achieved while reliably satisfying first-order optimality conditions, whereas

many competitors remain slightly non-stationary (yet trending toward stationary points) under the same equal-error budgets.

## H. Checking permutation equivalence between two factor matrices

Let $A$ and $B$ be two NMF algorithms returning factor matrices $(U_A, V_A)$ and $(U_B, V_B)$, respectively. We identify the subset of columns in $U_B$ that are scaled permutations of the columns in $U_A$ using the cosine similarity measure. Assuming $U_A$ and $U_B$ each have $r$ columns, we seek to find index sets

$$A_{\text{candidates}} \subseteq \{1, 2, \ldots, r\}, \qquad B_{\text{candidates}} \subseteq \{1, 2, \ldots, r\}$$

such that

$$\cos(U_A^{(i)}, U_B^{(P_i)}) = 1, \tag{24}$$

where $i \in A_{\text{candidates}}$ and $P_i \in B_{\text{candidates}}$. If (24) holds exactly, then we have identified the subset of columns in $U_A$ that map to columns in $U_B$. However, due to roundoff errors and numerical instability, Eq. (24) is rarely satisfied with equality. To address this, we relax the criterion to

$$\cos(U_A^{(i)}, U_B^{(P_i)}) \geq 1 - \epsilon,$$

where $\epsilon$ is a small tolerance parameter (we used $\epsilon = 0.05$ in our experiments). In addition to this relaxed criterion, we also perform additional cross cosine similarity checks to validate the identified subset.

Suppose we construct the following square submatrices:

$$M_{ij} = \cos(U_{\text{Asub}}^{(i)}, U_{\text{Asub}}^{(j)}),$$
$$N_{ij} = \cos(U_{\text{Bsub}}^{(i)}, U_{\text{Bsub}}^{(j)}),$$
$$S_{ij} = \cos(U_{\text{Asub}}^{(i)}, U_{\text{Bsub}}^{(j)}),$$

where

$$U_{\text{Asub}} = U_A[:, A_{\text{candidates}}],$$
$$U_{\text{Bsub}} = U_B[:, B_{\text{candidates}}].$$

Here $M$ and $N$ are self-similarity matrices and $S$ is the cross-similarity matrix. We sort the entries in each column of these matrices in descending order and perform the following steps for each column of $M$:

- Compare the entries of the column in $M$ with the corresponding entries of the columns in $N$ and $S$, rounding to one decimal place.

- If more than 90% of the entries in the column of $M$ match the corresponding entries in $N$ and $S$, we keep the associated indices in $A_{\text{candidates}}$ and $B_{\text{candidates}}$. Otherwise, we remove the indices from both sets.

After running these steps for all columns of $M$, the pruned sets $A_{\text{candidates}}$ and $B_{\text{candidates}}$ contain the indices of columns in $U_A$ and $U_B$ that are permutations of each other, based on cosine similarity.

### H.1. Exact factorization datasets and Trending Equivalence (TE)

The synthetic exact-NMF datasets provide us with a sandbox to determine such permutation equivalences, since the globally optimal factors are known (i.e., those used to construct the datasets) and are sparse by construction, lying on the boundaries of the positive orthant. In fact (see Fig. 1 (B)), the competing algorithms converged to global minima (i.e., attained a reconstruction error of zero for various sparsity levels), giving us the opportunity to explore how the factors obtained by the different algorithms are related. In Table 13, we tracked the percentage of the columns of the factor matrices that converged to a unique column of the ground-truth factors as a function of sparsity levels for the exact factorization scenario. It can be observed that when the algorithms converge to the global minimum, the factor matrices obtained from different algorithms are permuted versions of the ground-truth factors. Table 13 also provides an important observation regarding when an

*Table 13.* **Permutation results on synthetic data (exact factorization; sparsity sweep).** Each cell reports the quartet $(u_B, v_B, e_d, e_m)$, where $B$ is a competing algorithm and $A$ denotes the ground truth factors $(U_A, V_A)$. Here $u_B$ (%) is the fraction of columns of $U_B$ that have converged to a *unique* column of $U_A$, and $v_B$ (%) analogously for $V_B$. The reconstruction–error gap is $e_d = \|X - U_B V_B^\top\|_F - \|X - U_A V_A^\top\|_F$. The equivalence flag $e_m \in \{\mathbf{E}, \mathbf{TE}, \mathbf{NE}\}$ indicates (E) equivalent local minima (up to permutation/scaling), (TE) tending/asymptotically equivalent, or (NE) non-equivalent. Please see Fig. 7 for details on $e_m$.

| Sparsity | eNMF | HALS | NMF-ADMM | Grad-Mult | ALS |
|---|---|---|---|---|---|
| 0.1 | (98,95,8.7,TE) | (96,93,8.9,TE) | (8,5,37.3,TE) | (5,2,38.6,TE) | (10,7,36.5,TE) |
| 0.2 | **(100,100,0,E)** | (29,26,31,TE) | (13,9,47.2,TE) | (0,0,51.4,TE) | (18,16,46,TE) |
| 0.3 | **(100,100,0,E)** | **(100,100,0,E)** | (24,21,58.7,TE) | (23,22.5,62.2,TE) | (25,20,58,TE) |
| 0.4 | **(100,100,0,E)** | **(100,100,0,E)** | (91,85,9.8,TE) | (40,35,24.9,TE) | **(100,100,0,E)** |
| 0.5 | **(100,100,0,E)** | **(100,100,0,E)** | (96,92,2.7,TE) | (50,50,11.3,TE) | **(100,100,0,E)** |

| Sparsity | AO-ADMM | A-HALS | NeNMF | FPGM | Vavasis |
|---|---|---|---|---|---|
| 0.1 | (93,92,8.9,TE) | (85,80,9.3,TE) | (74,72,10.6,TE) | (49,46,21.7,TE) | (97,97,8.8,TE) |
| 0.2 | (35,33,31,TE) | (74,66,12.8,TE) | (61,57,15.4,TE) | (27,25,32.9,TE) | (99,98,1.6,TE) |
| 0.3 | **(100,100,0,E)** | **(100,100,0,E)** | **(100,100,0,E)** | **(100,100,0,E)** | **(100,100,0,E)** |
| 0.4 | **(100,100,0,E)** | **(100,100,0,E)** | **(100,100,0,E)** | **(100,100,0,E)** | **(100,100,0,E)** |
| 0.5 | **(100,100,0,E)** | **(100,100,0,E)** | **(100,100,0,E)** | **(100,100,0,E)** | **(100,100,0,E)** |

| Dataset | $r$ | eNMF vs HALS | eNMF vs NMF-ADMM | eNMF vs Grad-Mult | eNMF vs ALS | eNMF vs AO-ADMM | eNMF vs A-HALS | eNMF vs NeNMF | eNMF vs FPGM | eNMF vs Vavasis |
|---|---|---|---|---|---|---|---|---|---|---|
| | 5 | **(100,100,0,E)** | **(100,100,0,E)** | **(100,100,0,E)** | **(100,100,0,E)** | **(100,100,0,E)** | **(100,100,0,E)** | **(100,100,0,E)** | **(100,100,0,E)** | **(100,100,0,E)** |
| | 10 | **(100,100,0,E)** | (70,60,129.62,TE) | (80,80,18.96,TE) | **(100,100,0,E)** | (18,15,32.3,NE) | **(100,100,0,E)** | **(100,100,0,E)** | **(100,100,0,E)** | **(100,100,0,E)** |
| Face | 15 | (75,59,0.25,TE) | (0,0,207.8,NE) | (0,0,207.8,NE) | (75,59,0.25,TE) | **(100,100,0,E)** | **(100,100,0,E)** | **(100,100,0,E)** | (80,60,0.13,TE) | (80,60,0.15,TE) |
| | 20 | **(100,100,0,E)** | (10,5,187.93,NE) | **(100,100,0,E)** | **(100,100,0,E)** | **(100,100,0,E)** | **(100,100,0,E)** | **(100,100,0,E)** | **(100,100,0,E)** | **(100,100,0,E)** |
| | 25 | **(100,100,0,E)** | **(100,100,0,E)** | **(100,100,0,E)** | **(100,100,0,E)** | **(100,100,0,E)** | **(100,100,0,E)** | **(100,100,0,E)** | **(100,100,0,E)** | **(100,100,0,E)** |
| | 10 | (91,86,0.01,TE) | (70,70,0.09,TE) | (75,73,0.016,TE) | (93,89,0.0096,TE) | (97,97,0.004,TE) | (91,86,0.01,TE) | (97,97,0.004,TE) | (86,85.5,0.01,TE) | (89,87,0.009,TE) |
| | 20 | (92,90,0.008,TE) | (75,75,0.07,TE) | (81,75,0.03,TE) | (90,90,0.009,TE) | (97,97,0.004,TE) | (92,90,0.009,TE) | (96,95,0.005,TE) | (88,87.5,0.01,TE) | (89,89,0.008,TE) |
| Verb | 40 | (93,92,0.0004,TE) | (83,82,0.01,TE) | (85,84.75,0.01,TE) | (91,90,0.006,TE) | **(100,100,0,E)** | (93,93,0.0004,TE) | **(100,100,0,E)** | (90,89,0.007,TE) | (90,90,0.007,TE) |
| | 80 | **(100,100,0,E)** | (75.25,73.25,0.069,TE) | (75.25,73.25,0.075,TE) | (87,85.75,0.02,TE) | **(100,100,0,E)** | **(100,100,0,E)** | **(100,100,0,E)** | (94.5,92.25,0.01,TE) | (87,87,0.018,TE) |
| | 100 | **(100,100,0,E)** | (86,83,0.013,TE) | (86,83,0.01,TE) | (95,93,0.003,TE) | **(100,100,0,E)** | **(100,100,0,E)** | **(100,100,0,E)** | (98,96.65,0.00003,TE) | (86,83,0.01,TE) |
| | 10 | (60,50,35.47,TE) | (60,60,31.98,TE) | (60,55,35.81,TE) | (60,60,32.79,TE) | (70,60,27.97,TE) | (60,50,35.47,TE) | (60,60,20.16,TE) | (60,60,33.74,TE) | (60,55,36.27,TE) |
| | 20 | (75,70,7.29,TE) | (65,60,13.49,TE) | (65,60,13.51,TE) | (85,80,6.51,TE) | (85,80,6.39,TE) | (75,70,7.29,TE) | (80,80,7.6,TE) | (75,75,6.91,TE) | (65,65,12.69,TE) |
| Audio | 40 | **(100,100,0,E)** | (85,85,10.29,TE) | (80,75,14.99,TE) | **(100,100,0,E)** | **(100,100,0,E)** | **(100,100,0,E)** | **(100,100,0,E)** | **(100,100,0,E)** | **(100,100,0,E)** |
| | 80 | **(100,100,0,E)** | (90,80,2.67,TE) | (90,90,2.53,TE) | **(100,100,0,E)** | **(100,100,0,E)** | **(100,100,0,E)** | **(100,100,0,E)** | **(100,100,0,E)** | **(100,100,0,E)** |
| | 100 | **(100,100,0,E)** | **(100,100,0,E)** | **(100,100,0,E)** | **(100,100,0,E)** | **(100,100,0,E)** | **(100,100,0,E)** | **(100,100,0,E)** | **(100,100,0,E)** | **(100,100,0,E)** |

*Table 14.* Permutation Results on Real Datasets (Face, Verb, Audio). Each cell reports $(u_B, v_B, e_d, e_m)$; $A$ is the *eNMF* solution $(U_A, V_A)$ and methods are compared *vs.* eNMF across ranks $r$. Here $u_B$ and $v_B$ are the percentages of columns of $U_B$ and $V_B$ that converge to unique columns of $U_A$ and $V_A$; $e_d = \|X - U_B V_B^\top\|_F - \|X - U_A V_A^\top\|_F$; and $e_m \in \{\mathbf{E}, \mathbf{TE}, \mathbf{NE}\}$ summarizes equivalence via KKT verification (Table 12) and $e_d$. Please see Table 13 and Fig. 7 for details on $e_m$.

*algorithm trends towards a minimum that is equivalent to a ground-truth minimum or to a minimum achieved by another algorithm.* For example, at a sparsity level of 0.1 we find that, for the eNMF algorithm, 49 (47) columns out of 50 of its factor matrix $U$ ($V$) are permuted and scaled versions of the ground-truth factor matrices. That is, if $(U^\star, V^\star)$ is the globally optimal factor matrix pair, then there exists a permutation of the 50 columns of $U$ and $V$ obtained by eNMF such that for a subset of 45 columns we have $\cos(U_i, U^\star_{P(i)}) = 1 - \epsilon$ and $\cos(V_i, V^\star_{P(i)}) = 1 - \epsilon$, where $\epsilon \ll 1$ (we used $\epsilon = 0.05$ in our computations). In fact, if the permutation mapping is computed before eNMF reaches its minimum value (in this case, 8.7 instead of 0), then fewer columns of the factor matrices would have been permutations of the ground-truth factors. The AO-ADMM and HALS algorithms that reach a reconstruction error of 8.9 have fewer converged columns. For a sparsity level of 0.2, both AO-ADMM and HALS are further away and thus have fewer converged columns than eNMF, which reaches a global equivalent minimum and has perfect reconstruction.

This general phenomenon was also observed in other datasets where the globally optimal factor matrices are not known. Suppose algorithm $A$ reaches a local minimum (in almost all datasets we tested, eNMF reaches a local minimum as determined by the KKT conditions; see Eqn. 23). It is often the case that the landscape around the minimum is extremely flat along most directions, causing algorithm $B$ to only asymptotically approach an equivalent local minimum. In such cases, we expect a large fraction of the columns of $U_B$ and $V_B$ to be scaled permutations of the columns in $U_A$ and $V_A$. Moreover, as $B$ is run for more iterations, this fraction is expected to increase until the algorithm effectively stalls away from the minimum. We find that most of these algorithms do not progress, even after tens of thousands of additional iterations, and neither do they satisfy the KKT conditions. **In such cases, we say that $B$ is trending towards an equivalent (TE) local minimum as** $A$ (see Fig. 7).

| SNR Level | $r$ | eNMF vs HALS | eNMF vs NMF-ADMM | eNMF vs Grad-Mult | eNMF vs ALS | eNMF vs AO-ADMM | eNMF vs A-HALS | eNMF vs NeNMF | eNMF vs FPGM | eNMF vs Vavasis |
|---|---|---|---|---|---|---|---|---|---|---|
| | 50 | (91,90,14.77,TE) | (85,83,22.97,TE) | (82,82,25.55,TE) | (86,86,18.19,TE) | **(100,100,0,E)** | **(100,100,0,E)** | (93,93,11.57,TE) | (93,92,11.63,TE) | (93,91,13.02,TE) |
| | 100 | (90,90,14.98,TE) | (84,84,26.04,TE) | (82,81,30.09,TE) | (86,85,22.11,TE) | **(100,100,0,E)** | **(100,100,0,E)** | (93,91,12.18,TE) | (93,93,10.96,TE) | (93,92,11.65,TE) |
| 20dB | 200 | (90,90,16.21,TE) | (84,83,23.39,TE) | (80,80,28.76,TE) | (86,86,20.7,TE) | **(100,100,0,E)** | (98,97,6.82,TE) | (92,92,13.39,TE) | (91,91,14.02,TE) | (91,91,14.07,TE) |
| | 400 | (87,86,19.44,TE) | (82,82,27.5,TE) | (79,77,36.96,TE) | (84,84,22.8,TE) | **(100,100,0,E)** | (97,97,7.2,TE) | (92,92,10.51,TE) | (90,90,12.2,TE) | (90,89,12.53,TE) |
| | 500 | (85,85,24.73,TE) | (82,81,32.91,TE) | (78,77,39.51,TE) | (83,83,26.43,TE) | **(100,100,0,E)** | (95,95,8.66,TE) | (89,87,19.93,TE) | (89,88,18.85,TE) | (88,88,19.07,TE) |
| | 50 | **(100,100,0,E)** | **(100,100,0,E)** | (79,77,28.55,TE) | **(100,100,0,E)** | **(100,100,0,E)** | **(100,100,0,E)** | **(100,100,0,E)** | **(100,100,0,E)** | **(100,100,0,E)** |
| | 100 | **(100,100,0,E)** | **(100,100,0,E)** | (78,78,30.59,TE) | **(100,100,0,E)** | **(100,100,0,E)** | **(100,100,0,E)** | **(100,100,0,E)** | **(100,100,0,E)** | **(100,100,0,E)** |
| 40dB | 200 | **(100,100,0,E)** | (77,76,42.81,TE) | (75,74,56.36,TE) | **(100,100,0,E)** | **(100,100,0,E)** | **(100,100,0,E)** | **(100,100,0,E)** | **(100,100,0,E)** | **(100,100,0,E)** |
| | 400 | **(100,100,0,E)** | (76,76,39.4,TE) | (72,72,58.19,TE) | (76,76,39.76,TE) | **(100,100,0,E)** | **(100,100,0,E)** | **(100,100,0,E)** | **(100,100,0,E)** | **(100,100,0,E)** |
| | 500 | **(100,100,0,E)** | (74,72,50.2,TE) | (72,70,62.33,TE) | (72,71,53.69,TE) | **(100,100,0,E)** | **(100,100,0,E)** | **(100,100,0,E)** | **(100,100,0,E)** | **(100,100,0,E)** |
| | 50 | **(100,100,0,E)** | (95,95,8.81,TE) | (91,90,11.16,TE) | (96,95,8.6,TE) | **(100,100,0,E)** | **(100,100,0,E)** | **(100,100,0,E)** | **(100,100,0,E)** | **(100,100,0,E)** |
| | 100 | **(100,100,0,E)** | (94,93,11.38,TE) | (89,88,17.59,TE) | (94,94,11.28,TE) | **(100,100,0,E)** | **(100,100,0,E)** | **(100,100,0,E)** | **(100,100,0,E)** | **(100,100,0,E)** |
| 60dB | 200 | **(100,100,0,E)** | (90,90,19.07,TE) | (87,85,27.37,TE) | (92,92,17.63,TE) | **(100,100,0,E)** | **(100,100,0,E)** | **(100,100,0,E)** | **(100,100,0,E)** | **(100,100,0,E)** |
| | 400 | (97,97,1.99,TE) | (87,85,23.71,TE) | (81,81,31.02,TE) | (88,87,19.44,TE) | **(100,100,0,E)** | **(100,100,0,E)** | (99,99,1.02,TE) | (99,98,1.38,TE | (99,98,1.4,TE |
| | 500 | (96,95,3.49,TE) | (87,86,24.19,TE) | (85,84,26.94,TE) | (88,88,21.86,TE) | **(100,100,0,E)** | **(100,100,0,E)** | (97,96,2.72,TE) | (97,97,2.63,TE) | (97,97,2.51,TE) |
| | 50 | **(100,100,0,E)** | **(100,100,0,E)** | (97,96,2.69,TE) | **(100,100,0,E)** | **(100,100,0,E)** | **(100,100,0,E)** | **(100,100,0,E)** | **(100,100,0,E)** | **(100,100,0,E)** |
| | 100 | **(100,100,0,E)** | **(100,100,0,E)** | (86,85,31.39,TE) | **(100,100,0,E)** | **(100,100,0,E)** | **(100,100,0,E)** | **(100,100,0,E)** | **(100,100,0,E)** | **(100,100,0,E)** |
| 80dB | 200 | **(100,100,0,E)** | (95,95,6.36,TE) | (85,85,29.66,TE) | **(100,100,0,E)** | **(100,100,0,E)** | **(100,100,0,E)** | **(100,100,0,E)** | **(100,100,0,E)** | **(100,100,0,E)** |
| | 400 | **(100,100,0,E)** | (80,80,28.19,TE) | (80,79,33.59,TE) | (87,85,18.82,TE) | **(100,100,0,E)** | **(100,100,0,E)** | **(100,100,0,E)** | **(100,100,0,E)** | **(100,100,0,E)** |
| | 500 | **(100,100,0,E)** | (80,79,30.91,TE) | (77,75,46.1,TE) | (81,79,23.17,TE) | **(100,100,0,E)** | **(100,100,0,E)** | **(100,100,0,E)** | **(100,100,0,E)** | **(100,100,0,E)** |
| | 50 | **(100,100,0,E)** | **(100,100,0,E)** | **(100,100,0,E)** | **(100,100,0,E)** | **(100,100,0,E)** | **(100,100,0,E)** | **(100,100,0,E)** | **(100,100,0,E)** | **(100,100,0,E)** |
| | 100 | **(100,100,0,E)** | **(100,100,0,E)** | **(100,100,0,E)** | **(100,100,0,E)** | **(100,100,0,E)** | **(100,100,0,E)** | **(100,100,0,E)** | **(100,100,0,E)** | **(100,100,0,E)** |
| 100dB | 200 | **(100,100,0,E)** | **(100,100,0,E)** | (95,90,9.65,TE) | **(100,100,0,E)** | **(100,100,0,E)** | **(100,100,0,E)** | **(100,100,0,E)** | **(100,100,0,E)** | **(100,100,0,E)** |
| | 400 | **(100,100,0,E)** | (95,95,4.28,TE) | (87,87,13.29,TE) | (98,97,1.97,TE) | **(100,100,0,E)** | **(100,100,0,E)** | **(100,100,0,E)** | **(100,100,0,E)** | **(100,100,0,E)** |
| | 500 | **(100,100,0,E)** | (83,80,19.38,TE) | (80,80,21.72,TE) | (87,86,14.57,TE) | **(100,100,0,E)** | **(100,100,0,E)** | **(100,100,0,E)** | **(100,100,0,E)** | **(100,100,0,E)** |

*Table 15.* Permutation Results across five SNR levels. Each cell reports $(u_B, v_B, e_d, e_m)$; $A$ is the *eNMF* solution $(U_A, V_A)$ and methods are compared *vs. eNMF* across ranks $r$. Here $u_B$ and $v_B$ are the percentages of columns of $U_B$ and $V_B$ that converge to unique columns of $U_A$ and $V_A$; $e_d = \|X - U_B V_B^\top\|_F - \|X - U_A V_A^\top\|_F$; and $e_m \in \{\mathbf{E}, \mathbf{TE}, \mathbf{NE}\}$ summarizes equivalence via KKT verification and $e_d$.

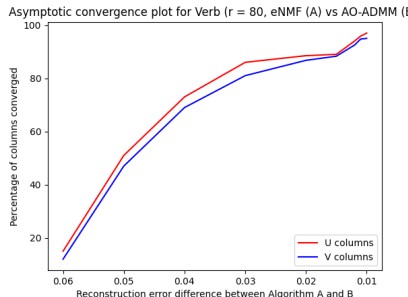

*Figure 7.* **Asymptotic/tending equivalence of factor matrices**: Percentage of columns in factor matrices returned by eNMF (Algorithm A) that are scaled permutations of the columns in factor matrices returned by AO-ADMM (Algorithm B) as a function of the reconstruction error difference between the algorithms. As the reconstruction error difference decreases, more columns in the factor matrices returned by AO-ADMM converge to scaled permutations of the columns in the factor matrices returned by eNMF. The percentage of converged columns approaches 100% as reconstruction error approaches zero, showing that the factor matrices tend asymptotically toward equivalent local minima.

## I. Computational Complexity Analysis of eNMF

The proposed eNMF algorithm is an iterative method for computing a feasible and accurate *approximate* nonnegative factorization. Accordingly, its computational complexity is characterized in terms of arithmetic operations per iteration, rather than worst-case decision complexity associated with the Exact NMF problem.

To make the derivation explicit, we decompose one outer iteration of eNMF into three main computational modules.

**(1) SVD initialization.** The eNMF algorithm initializes the factor matrices using a rank-$r$ truncated singular value decomposition of the input matrix $X$,

$$X \approx U_r \Sigma_r V_r^\top, \qquad U_r \in \mathbb{R}^{m \times r}, \ \Sigma_r \in \mathbb{R}^{r \times r}, \ V_r \in \mathbb{R}^{n \times r}.$$

The truncated SVD is computed using Krylov subspace methods such as Lanczos or Arnoldi iterations. The dominant computational cost arises from repeated matrix–vector products of the form $Xw$ and $X^\top w$, where $w \in \mathbb{R}^n$ or $w \in \mathbb{R}^m$. Each such product requires $O(mn)$ operations, and computing the leading $r$ singular components typically involves $O(r)$

such products, up to logarithmic factors and a small, method-dependent number of passes over the data. As a result, the leading-order complexity of the SVD initialization step is $O(mnr)$.

**2) Orthogonal rotation via ADMM.** The orthogonal rotation step (Algorithm 1) is solved using an ADMM scheme. Each ADMM iteration involves two dominant operations. First, Gram-type products and projections that couple the rotation variables with $U$ and $V$ incur a cost of $O((m+n)r^2)$. Second, solving a reduced $r \times r$ linear system or performing an eigendecomposition/SVD in the latent space costs $O(r^3)$. Consequently, one ADMM iteration requires

$$O\big((m+n)r^2 + r^3\big)$$

operations, and over $T_{\mathrm{admm}}$ iterations the total cost of the orthogonal rotation step is

$$O\Big(T_{\mathrm{admm}}\big((m+n)r^2 + r^3\big)\Big).$$

**3) Feasibility-enforcement loop.** The feasibility-enforcement stage (ASC) is repeated for $T_{\mathrm{asc}}$ outer iterations. Within each ASC iteration, three subroutines are invoked: PBCD updates for $U$ with $T^U_{\mathrm{pbcd}}$ iterations, PBCD updates for $V$ with $T^V_{\mathrm{pbcd}}$ iterations (Algorithm 2), and HALS refinements with $T_{\mathrm{hals}}$ iterations (Algorithm 3).

The NMF-type updates rely on the same set of matrix products, including the cross terms $XV \in \mathbb{R}^{m \times r}$, $X^\top U \in \mathbb{R}^{n \times r}$, and the Gram matrices $V^\top V \in \mathbb{R}^{r \times r}$, $U^\top U \in \mathbb{R}^{r \times r}$. Computing $XV$ or $X^\top U$ costs $O(mnr)$, while forming the Gram matrices and multiplying them with the factors contributes $O((m+n)r^2)$. Therefore, the dominant arithmetic cost associated with a single NMF update cycle is $O\big(mnr + (m+n)r^2\big)$.

Each inner iteration of these subroutines requires the same core NMF kernel described above, therefore a single ASC iteration costs

$$O\Big((T^U_{\mathrm{pbcd}} + T^V_{\mathrm{pbcd}} + T_{\mathrm{hals}})\big(mnr + (m+n)r^2\big)\Big),$$

and the total cost of the feasibility-enforcement stage is

$$O\Big(T_{\mathrm{asc}}(T^U_{\mathrm{pbcd}} + T^V_{\mathrm{pbcd}} + T_{\mathrm{hals}})\big(mnr + (m+n)r^2\big)\Big).$$

**Overall complexity.** Combining the costs of the core NMF kernel, the ADMM-based orthogonal rotation, and the feasibility-enforcement loop, the total computational complexity of eNMF can be expressed as

$$O\Big(mnr + T_{\mathrm{admm}}\big((m+n)r^2 + r^3\big)$$
$$+ T_{\mathrm{asc}}\big((T^U_{\mathrm{pbcd}} + T^V_{\mathrm{pbcd}} + T_{\mathrm{hals}})(mnr + (m+n)r^2)\big)\Big).$$

Under the low-rank assumption $r \ll \min\{m,n\}$, the overall cost is typically dominated by the repeated matrix–factor products of order $O(mnr)$ within the feasibility-enforcement loop, while the ADMM rotation step contributes lower-order terms unless either $r$ or $T_{\mathrm{admm}}$ is large.

## J. Additional Runtime Decomposition and Ablation Studies

*Table 16.* **Phase-wise runtime decomposition of eNMF on synthetic datasets under the equal-error protocol.** For each signal-to-noise ratio (SNR) and latent dimension $r$, we decompose the total wall-clock runtime of eNMF into three components: (i) truncated-SVD initialization, (ii) orthogonal rotation via Algorithm 1 (ADMM), and (iii) the subsequent feasibility-attainment plus final descent stage. **Key finding: on all synthetic settings, eNMF reaches the unconstrained global minimum entirely through SVD initialization and the exterior rotation step, with zero additional time spent in feasibility/descent.** Thus, for these datasets, the computational cost of eNMF is fully explained by the unconstrained low-rank factorization plus the orthogonal rotation toward the positive orthant. Runtime is reported in seconds.

| SNR | $r$ | SVD Initialization | Orthogonal Rotation with ADMM | Feasibility + Descent | Total Runtime |
|---|---|---|---|---|---|
| | 50 | 110 | 66 | 0 | 176 |
| | 100 | 126.4 | 71.97 | 0 | 198.4 |
| 20dB | 200 | 131.77 | 101.83 | 0 | 233.6 |
| | 400 | 152.06 | 129.54 | 0 | 281.6 |
| | 500 | 170.67 | 149.33 | 0 | 320 |
| | 50 | 71.99 | 39.99 | 0 | 111.98 |
| | 100 | 83.53 | 42.69 | 0 | 126.23 |
| 40dB | 200 | 85.48 | 63.15 | 0 | 148.63 |
| | 400 | 98.47 | 80.7 | 0 | 179.17 |
| | 500 | 111.17 | 92.44 | 0 | 203.6 |
| | 50 | 52.76 | 29.07 | 0 | 81.83 |
| | 100 | 57.78 | 34.45 | 0 | 92.24 |
| 60dB | 200 | 68.6 | 40.01 | 0 | 108.61 |
| | 400 | 73.16 | 57.77 | 0 | 130.93 |
| | 500 | 78.01 | 70.77 | 0 | 148.78 |
| | 50 | 35.97 | 22.51 | 0 | 58.48 |
| | 100 | 41.94 | 23.97 | 0 | 65.92 |
| 80dB | 200 | 48.08 | 29.54 | 0 | 77.62 |
| | 400 | 54.27 | 39.3 | 0 | 93.57 |
| | 500 | 59.25 | 47.08 | 0 | 106.33 |
| | 50 | 31.7 | 17.8 | 0 | 49.5 |
| | 100 | 34.52 | 21.28 | 0 | 55.8 |
| 100dB | 200 | 39.4 | 26.3 | 0 | 65.7 |
| | 400 | 45.52 | 33.68 | 0 | 79.2 |
| | 500 | 49.5 | 40.5 | 0 | 90 |

*Table 17.* **Phase-wise runtime decomposition of eNMF on real datasets under the equal-error protocol.** For each dataset and latent dimension $r$, we decompose the total wall-clock runtime of eNMF into three components: (i) truncated-SVD initialization, (ii) orthogonal rotation via Algorithm 1 (ADMM), and (iii) the subsequent feasibility-attainment plus final descent stage. **Key finding: on real datasets, most of eNMF's runtime is spent in SVD initialization and exterior rotation, while the feasibility/descent stage is consistently small and vanishes entirely for the Audio dataset at $r = 40, 80, 100$, where eNMF attains the unconstrained global minimum directly after rotation.** More broadly, even when the rotated SVD point does not immediately lie in the nonnegative orthant, the additional constrained correction is lightweight relative to the total runtime, showing that the practical advantage of eNMF comes primarily from starting at the unconstrained optimum and approaching feasibility from the exterior rather than spending most of the computation in interior descent. Runtime is reported in seconds.

| Dataset | $r$ | SVD Initialization | Orthogonal Rotation with ADMM | Feasibility + Descent | Total Runtime |
|---|---|---|---|---|---|
| | 5 | 7.25 | 1.67 | 0.88 | 9.8 |
| | 10 | 94.22 | 29.2 | 9.29 | 132.7 |
| Face | 15 | 134.05 | 58.9 | 10.16 | 203.1 |
| | 20 | 160.16 | 73.92 | 12.32 | 246.4 |
| | 25 | 190.2 | 112.25 | 9.35 | 311.8 |
| | 10 | 3.72 | 0.93 | 0.52 | 5.17 |
| | 20 | 3.45 | 1.08 | 0.39 | 4.93 |
| Verb | 40 | 5.78 | 2.45 | 0.53 | 8.76 |
| | 80 | 7.98 | 3.68 | 0.61 | 12.28 |
| | 100 | 9.38 | 4.55 | 0.73 | 14.66 |
| | 10 | 20.02 | 6.67 | 1.11 | 27.81 |
| | 20 | 37.01 | 15.78 | 1.63 | 54.43 |
| Audio | 40 | 42.23 | 19.87 | 0 | 62.11 |
| | 80 | 44.58 | 25.71 | 0 | 70.29 |
| | 100 | 60.33 | 47.38 | 0 | 107.74 |

*Table 18.* **Orthogonality and convergence behavior of the ADMM rotation step on synthetic datasets.** For each signal-to-noise ratio (SNR) and latent dimension $r$, we report (i) the orthogonality error of the matrix $R$ returned by Algorithm 1, measured by $\|R^\top R - I\|_F^2$, and (ii) the number of ADMM iterations required for convergence under our stopping rule. **Key finding: across all synthetic settings, Algorithm 1 returns an exactly orthogonal rotation matrix and converges in only 1–5 ADMM steps.** This shows that, although the rotation subproblem is nonconvex, the ADMM procedure is numerically extremely stable in practice on the synthetic benchmarks considered here. In particular, the returned matrix satisfies the orthogonality constraint to machine precision in every case, and the required iteration count remains very small even as the latent dimension increases to $r = 500$. These results support our use of Algorithm 1 as a lightweight and reliable rotation stage within the overall eNMF pipeline.

| SNR | $r$ | $\|R^T R - I\|_F^2$ | # ADMM steps |
|---|---|---|---|
| | 50 | 0 | 3 |
| | 100 | 0 | 2 |
| 20dB | 200 | 0 | 3 |
| | 400 | 0 | 5 |
| | 500 | 0 | 5 |
| | 50 | 0 | 2 |
| | 100 | 0 | 2 |
| 40dB | 200 | 0 | 3 |
| | 400 | 0 | 4 |
| | 500 | 0 | 4 |
| | 50 | 0 | 2 |
| | 100 | 0 | 2 |
| 60dB | 200 | 0 | 2 |
| | 400 | 0 | 3 |
| | 500 | 0 | 4 |
| | 50 | 0 | 2 |
| | 100 | 0 | 2 |
| 80dB | 200 | 0 | 2 |
| | 400 | 0 | 3 |
| | 500 | 0 | 3 |
| | 50 | 0 | 2 |
| | 100 | 0 | 1 |
| 100dB | 200 | 0 | 1 |
| | 400 | 0 | 2 |
| | 500 | 0 | 2 |

*Table 19.* **Orthogonality and convergence behavior of the ADMM rotation step on real datasets.** For each dataset and latent dimension $r$, we report (i) the orthogonality error of the matrix $R$ returned by Algorithm 1, measured by $\|R^\top R - I\|_F^2$, and (ii) the number of ADMM iterations required for convergence under our stopping rule. **Key finding: across all real-data settings, Algorithm 1 returns an almost perfectly orthogonal rotation matrix and converges in only 2–7 ADMM steps.** Although the rotation subproblem is nonconvex, the ADMM procedure is numerically very stable in practice: the returned matrix satisfies the orthogonality constraint to high precision, with $\|R^\top R - I\|_F^2$ ranging from $10^{-5}$ to $10^{-14}$ and reaching exactly 0 for several Audio settings. The iteration count remains very small across Face, Verb, and Audio, showing that the rotation stage is lightweight even on heterogeneous real datasets and larger latent dimensions. These results support viewing Algorithm 1 as a practical and reliable component of the overall eNMF pipeline.

| Dataset | $r$ | $\|R^T R - I\|_F^2$ | # ADMM steps |
|---|---|---|---|
| | 5 | $10^{-5}$ | 2 |
| | 10 | $10^{-5}$ | 5 |
| Face | 15 | $10^{-9}$ | 6 |
| | 20 | $10^{-9}$ | 6 |
| | 25 | $10^{-7}$ | 7 |
| | 10 | $10^{-6}$ | 2 |
| | 20 | $10^{-6}$ | 2 |
| Verb | 40 | $10^{-8}$ | 4 |
| | 80 | $10^{-8}$ | 4 |
| | 100 | $10^{-14}$ | 6 |
| | 10 | $10^{-6}$ | 4 |
| | 20 | $10^{-10}$ | 4 |
| Audio | 40 | 0 | 2 |
| | 80 | 0 | 3 |
| | 100 | 0 | 3 |

*Table 20.* **Ablation of the feasibility-attainment stage starting from the same rotated SVD initialization on real datasets under the equal-error protocol.** For each dataset and latent dimension $r$, we compare five post-rotation strategies applied after Algorithm 1: (i) our proposed feasibility-attainment stage followed by HALS descent, (ii) direct projection onto the nonnegative orthant followed by HALS descent, (iii) direct projection followed by Grad-Mult descent, (iv) direct projection followed by standard gradient descent, and (v) our feasibility-attainment stage followed by gradient descent. **Key finding: starting from the same rotated SVD point, our feasibility-attainment strategy with HALS descent is consistently the fastest, while the feasibility-based variants also outperform their projection-based counterparts, showing that the ascent/PBCD stage preserves useful low-rank structure better than a forceful projection.** Across Face, Verb, and lower-rank Audio settings, our method yields the lowest runtime in every case. For Audio at $r = 40, 80, 100$, all methods require zero additional runtime because the rotated solution already reaches the target solution directly after the ADMM rotation step. Runtime is reported in seconds.

| Dataset | $r$ | Feasibility + HALS Descent (Ours) | Projection + HALS Descent | Projection + Grad-Mult Descent | Projection + Gradient Descent | Feasibility + Gradient Descent |
|---|---|---|---|---|---|---|
| | 5 | 0.88 | 1.003 | 1.06 | 1.14 | 0.98 |
| | 10 | 9.29 | 10.5 | 11.15 | 11.74 | 10.21 |
| Face | 15 | 10.16 | 11.38 | 12.09 | 12.62 | 10.77 |
| | 20 | 12.32 | 14.04 | 14.53 | 15.57 | 13.16 |
| | 25 | 9.35 | 10.29 | 10.84 | 12.03 | 10.08 |
| | 10 | 0.52 | 0.598 | 0.63 | 0.67 | 0.58 |
| | 20 | 0.39 | 0.45 | 0.47 | 0.51 | 0.43 |
| Verb | 40 | 0.53 | 0.6 | 0.63 | 0.68 | 0.59 |
| | 80 | 0.61 | 0.7 | 0.72 | 0.79 | 0.67 |
| | 100 | 0.73 | 0.79 | 0.86 | 0.94 | 0.75 |
| | 10 | 1.11 | 1.27 | 1.33 | 1.53 | 1.18 |
| | 20 | 1.63 | 1.75 | 1.84 | 2.07 | 1.68 |
| Audio | 40 | 0 | 0 | 0 | 0 | 0 |
| | 80 | 0 | 0 | 0 | 0 | 0 |
| | 100 | 0 | 0 | 0 | 0 | 0 |

*Table 21.* **Ablation of the feasibility-attainment stage: optimal row-wise step size versus fixed step size on real datasets under the equal-error protocol.** For each dataset and latent dimension $r$, we compare our feasibility-attainment stage using the closed-form optimal row-wise step sizes from Eqs. (10)–(11), followed by HALS descent, against an otherwise identical variant using a fixed step size. **Key finding: the optimal-step-size version is consistently faster across all nontrivial real-data settings, showing that the closed-form row-wise updates improve the efficiency of feasibility attainment without introducing additional tuning.** Across Face, Verb, and lower-rank Audio, the optimal-step-size variant yields the lowest runtime in every case. For Audio at $r = 40, 80, 100$, both methods require zero additional runtime because the rotated solution already reaches the target solution directly after the ADMM rotation step. Runtime is reported in seconds.

| Dataset | $r$ | Feasibility with optimal step size + HALS Descent (Ours) | Feasibility with fixed step size + HALS Descent |
|---|---|---|---|
| | 5 | 0.88 | 1.06 |
| | 10 | 9.29 | 11.33 |
| Face | 15 | 10.16 | 12.29 |
| | 20 | 12.32 | 15.64 |
| | 25 | 9.35 | 12.06 |
| | 10 | 0.52 | 0.63 |
| | 20 | 0.39 | 0.49 |
| Verb | 40 | 0.53 | 0.64 |
| | 80 | 0.61 | 0.73 |
| | 100 | 0.73 | 0.91 |
| | 10 | 1.11 | 1.43 |
| | 20 | 1.63 | 2.05 |
| Audio | 40 | 0 | 0 |
| | 80 | 0 | 0 |
| | 100 | 0 | 0 |

*Table 22.* **Equal-error runtime comparison between eNMF and ANLS-BPP on real datasets.** For each dataset and latent dimension $r$, we report the wall-clock time required by eNMF and ANLS-BPP to reach the same reconstruction-error target, namely the error attained by eNMF at convergence. **Key finding: eNMF consistently outperforms the strong ANLS-BPP baseline across all tested real datasets and latent dimensions, showing that the exterior initialization strategy remains advantageous even against a competitive NNLS-based solver.** Across Face, Verb, and Audio, eNMF achieves the target error faster in every setting. The advantage is especially pronounced on Audio at higher latent dimensions, where eNMF attains the unconstrained global minimum for $r = 40, 80, 100$, while ANLS-BPP still requires substantially more runtime to match the same error target. Runtime is reported in seconds.

| Dataset | $r$ | eNMF Total Runtime | ANLS-BPP Total Runtime |
|---------|-----|--------------------|------------------------|
| Face | 5 | 9.8 | 11.41 |
|  | 10 | 132.7 | 147.19 |
|  | 15 | 203.1 | 225.66 |
|  | 20 | 246.4 | 279.37 |
|  | 25 | 311.8 | 336.9 |
| Verb | 10 | 5.17 | 5.86 |
|  | 20 | 4.93 | 5.58 |
|  | 40 | 8.76 | 9.83 |
|  | 80 | 12.28 | 15.02 |
|  | 100 | 14.66 | 17.06 |
| Audio | 10 | 27.81 | 34.75 |
|  | 20 | 54.43 | 66.48 |
|  | 40 | 62.11 | 73.61 |
|  | 80 | 70.29 | 86.23 |
|  | 100 | 107.74 | 141.3 |

*Table 23.* **Equal-error runtime comparison for eNMF with exact truncated SVD versus randomized SVD initialization on real datasets.** For each dataset and latent dimension $r$, we report the total wall-clock runtime of eNMF when initialized with the exact truncated SVD and when initialized with a randomized SVD approximation. **Key finding: randomized SVD yields essentially identical end-to-end runtime to exact truncated SVD on these benchmarks, indicating that eNMF is robust to replacing the exact SVD with a high-quality approximate low-rank initializer.** Across Face, Verb, and Audio, the total runtimes are nearly indistinguishable, with only negligible differences at larger ranks. This suggests that the later rotation, feasibility, and descent stages behave almost identically under the two initializations, and supports the view that eNMF depends primarily on the quality of the low-rank subspace rather than on exact computation of the truncated SVD itself. For the Audio dataset at $r = 40, 80, 100$, where eNMF attains the unconstrained global minimum, the randomized-SVD initialization preserves the same practical behavior while slightly reducing total runtime at higher ranks. Runtime is reported in seconds.

| Dataset | $r$ | eNMF Total Runtime (exact truncated SVD) | eNMF Total Runtime (Randomized SVD) |
|---------|-----|------------------------------------------|-------------------------------------|
| Face | 5 | 9.8 | 9.8 |
|  | 10 | 132.7 | 132.7 |
|  | 15 | 203.1 | 203.1 |
|  | 20 | 246.4 | 246.4 |
|  | 25 | 311.8 | 312 |
| Verb | 10 | 5.17 | 5.17 |
|  | 20 | 4.93 | 4.93 |
|  | 40 | 8.76 | 8.78 |
|  | 80 | 12.28 | 12.29 |
|  | 100 | 14.66 | 14.69 |
| Audio | 10 | 27.81 | 27.81 |
|  | 20 | 54.43 | 54.43 |
|  | 40 | 62.11 | 62.07 |
|  | 80 | 70.29 | 69.92 |
|  | 100 | 107.74 | 106.38 |

*Table 24.* **Equal-error runtime comparison on the ultra-large Reuters dataset** ($804{,}414 \times 47{,}236$, $99.84\%$ **sparsity**). We compare the wall-clock time required by eNMF, ANLS-BPP, HALS, and A-HALS to reach the same reconstruction-error target on an extremely large and highly sparse text dataset. **Key finding: even in this ultra-large sparse regime, eNMF remains the fastest method across all tested latent dimensions, showing that the SVD-based exterior initialization does not negate its wall-clock advantage relative to lightweight interior solvers.** Across all ranks $r = 10, 20, 40, 80, 100$, eNMF consistently outperforms the strong NNLS-based baseline ANLS-BPP as well as HALS and A-HALS. The gap widens with increasing latent dimension, indicating favorable scaling of the overall eNMF pipeline even when the underlying matrix is extremely sparse and high-dimensional. Runtime is reported in seconds.

| $r$ | eNMF | ANLS-BPP | HALS | A-HALS |
|-----|------|----------|------|--------|
| 10 | 41.72 | 44.22 | 47.56 | 45.47 |
| 20 | 57.66 | 62.27 | 67.46 | 64.01 |
| 40 | 83.04 | 88.85 | 96.33 | 91.34 |
| 80 | 126.19 | 137.55 | 145.12 | 141.33 |
| 100 | 152.81 | 169.62 | 181.84 | 172.68 |

*Table 25.* **Equal-time constrained relative reconstruction errors on real datasets for different post-rotation strategies.** Starting from the same rotated SVD initialization, we compare five post-rotation variants under an identical wall-clock budget: (i) our proposed feasibility-attainment stage followed by HALS descent, (ii) direct projection onto the nonnegative orthant followed by HALS descent, (iii) direct projection followed by Grad-Mult descent, (iv) direct projection followed by standard gradient descent, and (v) our feasibility-attainment stage followed by gradient descent. Each entry reports the reconstruction error relative to that obtained by our method, i.e., the ratio RE(method)/RE(ours), so the value for our method is always 1.00 and lower values are better. **Key finding: under the same time budget, our feasibility + HALS strategy consistently achieves the lowest reconstruction error across all datasets and latent dimensions, while projection-based alternatives incur substantially larger error, especially on Audio and at higher ranks.** This shows that the advantage of the proposed feasibility-attainment stage is not only computational efficiency, but also superior preservation and refinement of the rotated low-rank solution under practically relevant equal-time constraints. In particular, the gap widens for harder settings, indicating that forceful projection destroys useful structure that our ascent/PBCD stage is able to retain. For the Audio dataset at $r = 40, 80, 100$, the rotated SVD solution already attains the unconstrained global minimum; as a result, all post-rotation strategies start from the same optimal point (no need for any post rotation phase) and therefore have relative error ratio equal to 1.00.

| Dataset | $r$ | Feasibility + HALS Descent (Ours) | Projection + HALS Descent | Projection + Grad-Mult Descent | Projection + Gradient Descent | Feasibility + Gradient Descent |
|---|---|---|---|---|---|---|
| Face | 5 | **1.00** | 1.07 | 1.1 | 1.14 | 1.05 |
| | 10 | **1.00** | 1.08 | 1.12 | 1.16 | 1.06 |
| | 15 | **1.00** | 1.1 | 1.12 | 1.16 | 1.08 |
| | 20 | **1.00** | 1.12 | 1.15 | 1.19 | 1.12 |
| | 25 | **1.00** | 1.13 | 1.16 | 1.24 | 1.12 |
| Verb | 10 | **1.00** | 1.07 | 1.16 | 1.17 | 1.07 |
| | 20 | **1.00** | 1.11 | 1.18 | 1.17 | 1.1 |
| | 40 | **1.00** | 1.13 | 1.19 | 1.22 | 1.11 |
| | 80 | **1.00** | 1.14 | 1.22 | 1.27 | 1.14 |
| | 100 | **1.00** | 1.16 | 1.25 | 1.29 | 1.15 |
| Audio | 10 | **1.00** | 1.04 | 1.15 | 1.23 | 1.07 |
| | 20 | **1.00** | 1.12 | 1.17 | 1.27 | 1.03 |
| | 40 | **1.00** | 1.00 | 1.00 | 1.00 | 1.00 |
| | 80 | **1.00** | 1.00 | 1.00 | 1.00 | 1.00 |
| | 100 | **1.00** | 1.00 | 1.00 | 1.00 | 1.00 |

