# OpenReview forum: "An Exterior Method for Nonnegative Matrix Factorization"
_ICML.cc/2026/Conference — ICML 2026 regular_

### Official Review · Reviewer_CGPr · 2026-02-25

**Soundness:** 3
**Presentation:** 4
**Significance:** 3
**Originality:** 3
**Overall Recommendation:** 5
**Confidence:** 3

**Summary:**

The authors revisit the non-negative matrix factorization problem (NMF) by using an exterior method instead of interior ones. The fact that the method is exterior comes from the fact that the initialization is made by an SVD approximation of the original matrix, which then is updated in three stages: rotation (to get close to the non-negative orthant), then penalty (to get inside the feasibility region) and finally gradient descent. The authors provide empirical evidence of the improvements made by their algorithm.

**Compliance With Llm Reviewing Policy:**

Affirmed.

**Key Questions For Authors:**

1. In Algorithm 1, page 4, what is the stopping rule for the updates ?
2. More generally, how does the stopping rules in each part of the algorithm influence the accuracy and run-time of the algorithm ? I think this is not well-explained in the main text.
3. Are there "adversarial" cases where interior algorithms could out-perform eNMF ?

**Limitations:**

The authors comment the limitations of their work.

**Strengths And Weaknesses:**

Strengths: Overall the work is sound and original, as it improves over classic benchmarks and revisits a classic problem bringing new ideas.
1. The work revisits the NMF through a novel approach and a simple algorithm, which is original both from the fact that is exterior and the fact that is simple.
2. The algorithm is technically sound and intuitive, and its presentation is very well-written, which makes it easier to follow.

Weaknesses:
1. The authors discuss related work on NMF, but as a non-expert in this area, I wonder whether exterior methods have been used before in similar problems.
2. The last appendix, deriving the run-time complexity of the eNMF algorithm proposed by the authors is barely referenced in the main text, and this discussion seems fundamental, as the authors claim that the eNMF algorithm has a lower run-time than the algorithms using interior methods.

---

> ### Author Rebuttal · Authors · 2026-03-31
>
> We thank the reviewer for recognizing the originality and soundness of our work, particularly its novel yet simple exterior approach to NMF, along with the clarity and intuitive presentation of the algorithm. The key questions are addressed below. All tables referred in this response can be found at this link https://anonymous.4open.science/r/ICML-8F26/README.md
>
> ---
>
> Q1: In Algorithm 1, what is the stopping rule for the updates ?
>
> A:  We agree that the stopping rule for Algorithm 1 should be stated explicitly in the manuscript. In our implementation, the ADMM updates in Algorithm 1 are terminated when both the primal residual and the dual residual fall below a prescribed tolerance. Specifically, we monitor the norm of the primal residual associated with the consensus constraint Z = WR and the norm of the corresponding dual residual, and stop when both are below a fixed threshold (we use $10^{-4}$). In addition, we impose a maximum number of iterations (max_iter = 4000) as a safeguard.
> We also note that, in practice, Algorithm 1 converges far earlier than this maximum limit. As shown in our added convergence table(Table 3,4), the ADMM rotation step requires only a very small number of iterations across all datasets and latent dimensions. On the synthetic datasets, convergence is reached in only 1-5 ADMM steps across all tested SNR values and ranks. On the real datasets, convergence occurs in only 2-7 steps across Face, Verb, and Audio, while the returned matrix remains orthogonal. Thus, although we include max_iter = 4000 as a generic safeguard, the actual number of iterations required in practice is very small in all scenarios we tested.
> We will revise Algorithm 1 and the implementation details section to state this stopping rule explicitly and to reference the convergence table showing the small number of ADMM iterations required across datasets and latent dimensions.
>
> ---
>
> Q2: More generally, how does the stopping rules in each part of the algorithm influence the accuracy and run-time of the algorithm ? I think this is not well-explained in the main text.
>
> A: eNMF has two main parts: Algorithm 1 that involves rotation to the nearest point to the positive orthant using ADMM, and then the Ascent to reach feasibility, followed by HALS until convergence to the KKT conditions.  For the ADMM part as mentioned in the preceding answer, it converges in single-digit number of iterations; hence, it is extremely stable and the stopping criteria are met very fast. For the other parts, it is the usual KKT condition criteria and they are met by our eNMF algorithm for all the numerical examples we studied. As argued in our paper, it is because we  approach the positive orthant from the outside.
>
> ---
>
> Q3: Are there "adversarial" cases where interior algorithms could out-perform eNMF ?
>
> A: Yes, in principle there can be adversarial cases in which an interior NMF algorithm outperforms eNMF. Our method is designed to exploit the geometry of the unconstrained low-rank optimum by starting from the truncated-SVD solution and searching for a rotation toward the positive orthant. Therefore, eNMF is expected to be most effective when this exterior route places the iterate close to a high-quality feasible NMF solution. In contrast, if the SVD-based manifold is poorly aligned with the relevant nonnegative basin, then the advantage of the exterior initialization can diminish, and an interior method initialized directly inside the orthant could in principle converge faster or to a better local minimum.
> That said, our experiments suggest that such cases are uncommon in the regimes we tested. Across the synthetic datasets and across the real Face, Verb, and Audio benchmarks, eNMF consistently achieved the best equal-time reconstruction error and the best equal-error runtime among all compared methods. Moreover, in the large-scale equivalence analysis reported in the paper, distinct non-equivalent local minima occurred in only a small minority of cases, while in the overwhelming majority of settings the competing algorithms trended toward solutions that were geometrically equivalent, or nearly equivalent, to the eNMF solution. This suggests that, in practice, the main advantage of eNMF is not necessarily that it reaches a fundamentally different solution in most cases, but that it reaches a high-quality solution much faster by approaching from the exterior.
> A more challenging regime for eNMF would therefore be one in which the rotated SVD point remains far from the useful feasible region and the final feasibility/descent stage becomes dominant. In that situation, the benefit of the exterior approach would be reduced, and an interior method could potentially be competitive or superior. We will clarify in the revision that eNMF is not claimed to dominate all possible NMF instances, but rather to provide a consistently advantageous strategy when the unconstrained optimum offers a strong geometric guide toward the nonnegative solution.

---

> > ### Author Rebuttal · Reviewer_CGPr · 2026-04-01
> >
> > My questions have resolved.

---

### Official Review · Reviewer_trzz · 2026-03-12

**Soundness:** 2
**Presentation:** 2
**Significance:** 3
**Originality:** 3
**Overall Recommendation:** 4
**Confidence:** 4

**Summary:**

The manuscript proposes a novel algorithm for non-negative matrix factorization (NMF). The algorithm, dubbed eNMF, uses an initialization step based on an ADMM on a sparsity-promoting orthogonal Procrustes problem, followed by a coordinate descent-type approach. The algorithm's design carefully tries to take into ambiguities that exist within matrix factorizations, which is interesting. The eNMF algorithm is then experimentally explored quite extensively through comparisons to other state-of-the-art NMF algorithms regarding reconstruction error, runtime and downstream task performance on synthetic data with exact non-negative matrix factors as well as standard datasets coming from NMF applications (verb, audio and face dataset). A significant theoretical convergence analysis is not provided. A Python repository with limited functionality of code implementations of the algorithms considered is provided.

**Compliance With Llm Reviewing Policy:**

Affirmed.

**Final Justification:**

The discussion period addressed several of my concerns, which is why I updated my rating.

**Key Questions For Authors:**

- What is M_E in Algorithm 2 and how does it fit into the algorithm's derivation? It seems to be set to the matrix of ones in Algorithm 3, but I do not understand this notation without reading the appendix.
 - What choices are made for $R^0$, $Y^0$ and $\rho$ in the eNMF algorithm of Algorithm 3 (e.g., as used in the numerical experiments) and why?
 - Can you provide some ablations that justify your design choices (see weakness pointed out above)?
 - Can you provide comparisons to ANLS-BPP and scikit-learn-style coordinate descent for some of your numerical experiments?
 - Can you provide a breakdown of the different phases of your algorithm in terms of runtime (initialization, mid-phase, HALS at the end)?

**Limitations:**

Yes.

**Strengths And Weaknesses:**

I consider the following aspects as the key **strengths** of the paper:

- The key idea of eNMF, which involves the orthogonal Procrustes initialization with non-negativity penality, is to the best of my knowledge novel and interesting for tackling NMF problems.
- Related to this, the geometric viewpoint of the paper that considers equivalence classes of rotated factor matrices is a credible ingredient for the novel algorithm design, and might be of general interest in the NMF literature (if novel).
- The numerical experiments suggest that eNMF might be a top NMF solver, especially regarding overall runtime, if used correctly.

On the other hand, I need to point out several **weaknesses**:
- The update equations of (6) and (7) for the negative entries derived from the penalty method of 4.2 seems to not follow a typical gradient descent, despite this being claimed in line 173: In fact, (6) and (7) should also have a term related to the residual error function f, similar to what appears in the update equations (8) and (9).
  This issue needs to be addressed. If this "modified gradient descent" leads to better results than standard GD, this should be at least explored in an ablation study.
- A key selling point of the eNMF methodology and design principle of the numerical evaluation is its computational advantage in terms of run time, as in most cases, all algorithms converge to equivalent solutions. To provide sufficient empirical proof of this advantage, an evaluation with very efficient (but theoretically maybe less proven) NMF algorithms such as ANLS-BPP from [Kim and Park 2011, SIAM J. Sci. Comp] or a fair variant of scikit learn's coordinate descent NMF solver would be beneficial.
- Key parameter choices in the framework are not elaborated on. What are $\delta_u$, $\delta_v$, the tolerance in the ADMM of Algorithm 1, the initialization of ADMM of Algorithm 1 ($R^0$, $Y^0$ and $\rho$)?
- Overall, reasonable ablations for the design elements of the eNMF algorithm are missing: In particular, it is not shown whether Algorithm 1 is actually necessary in practice, why the penalty method is solved by the mentioned "modified (projected) gradient descent" rather than a standard projected GD algorithm, and how sensitive the algorithm is to some of the hyperparameters such as rank under or overspecification, ADMM iterations, etc. The comparison between "F-eNMF" and eNMF of Figure 5 is not really explained, but seems to assume to use Algorithm 1 in both cases. The benefit of using the optimal stepsize of (10) and (11) vs. a fixed step size is not explored in an ablation either.
- The presentation of the paper focusses on an outline of the novel algorithmic ideas and its relation to certain optimization formulations, as well as an explanation of the experimental setup. However, the presentation is sometimes convoluted or misleading: In Section 4.1, where the initialization part of eNMF framework are explained, the function $h$, which is key in the definition of the underlying optimzation modelling, is only introduced a few paragraphs after its first mention. The optimization formulation (16) from the appendix is referred to before (4) and, which leads to challenges for a reader trying to understand the methodology.
- I was not able to run the code that the authors provided to reproduce the experiments, as some of the datasets "experiment_scripts_exacts.py" and "run_experiment_demo.py" were missing and no guidance how to obtain them was provided in the readme files (in particular, Dataset/verb/right_matrix.npy and Dataset/exact_dataset/exacts_500_400_50_0.1.npy. For this reason, I was not able to verify the experimental claims made in the paper, which limits the experimental reproducibility. That being said, the experimental claims are somewhat plausible.

Overall, I consider the problem of NMF as a significant problem in machine learning with many application, and if the authors indeed contribute to a faster solver, this is valuable for the community. While I might be not aware of all significant algorithms in the NMF literature, I consider the idea of the method generally rather original. If the weaknesses pointed out above (which touch mostly upon soundness and presentation) can be addressed, I would be willing to adapt my rating of the submission.

---

> ### Author Rebuttal · Authors · 2026-03-31
>
> We thank the reviewer for recognizing the novelty of our framework and our contribution in introducing a geometric viewpoint based on equivalence classes. The key questions are addressed below. All tables referred in this response can be found at this link https://anonymous.4open.science/r/ICML-8F26/README.md
>
> Q1: What is $M_E$ in Algorithm 2 and how does it fit into the algorithm's derivation?
>
> A:  $M_E$ is a binary observation mask indicating which entries of data matrix $X$ contribute to the reconstruction loss. In Algorithm 2, the residual is written as $M_E \circ (UV^\top - X)$ so that only observed entries affect the updates. In Algorithm 3 (NMF), all entries are observed, so $M_E$ is the all-ones matrix. For matrix completion (Appendix C), $M_E$ encodes the observation pattern, with missing entries excluded.
>
> ---
>
> Q2: What choices are made for $R^{(0)}, Y^{(0)}$ and $\rho$ in Algorithm 3 and why?
>
> A:  We use fixed settings across all experiments: $\rho=5$, $Y^{(0)}=\mathbf{1}$, and $R^{(0)}=I$. We did not need any data-specific fine tuning of these hyper parameters, as one set worked robustly for all the datasets.
>
> - $\rho=5$ (ADMM penalty) ensures fast, stable convergence (1–7 iterations) with near-perfect orthogonality ($\|R^\top R - I\|_F^2 \approx 10^{-5}$–$10^{-14}$). It mainly affects convergence speed, not the final solution (Tables 3,4).
> - $Y^{(0)}=\mathbf{1}$ is a fixed initialization; as a dual variable, it primarily influences early iterations and showed stable convergence across all settings.
> - $R^{(0)}=I$ starts from the unrotated SVD solution, providing a neutral, parameter-free initialization that lets ADMM learn the rotation directly.
>
> We will clarify these choices in the revision.
>
> ---
>
> Q3: Can you provide ablations to justify design choices?
>
>
> Q3.1: Updates (6)–(7) do not follow standard GD.
>
> A:  We agree the wording was imprecise. Eqs. (6)–(7) are not full gradient-descent updates; they are penalty-dominated approximations applied only to negative entries when the penalty term outweighs the reconstruction gradient. The exact gradient would include the residual term, as noted. We will clarify this.
>
> We also conducted an ablation starting from the same rotated SVD point: (i) Projection + HALS and (ii) Projection + Grad-Mult. Our feasibility + HALS stage is consistently faster across real datasets, while all methods require no extra time when the rotated solution is already feasible. Overall, the feasibility step is a lightweight penalty-driven correction—not an exact GD solver—and is empirically more effective than projection-based alternatives. (Table 5)
>
>
> ---
>
> Q3.2: Optimal vs fixed step size?
>
> A:  We conducted an ablation comparing optimal row-wise step sizes (Eqs. 10–11) versus a fixed step size to reach feasibility; then followed it by HALS as done in our eNMF (Table 6) . The optimal-step-size version is consistently faster across real datasets (Face, Verb, Audio at low ranks), while both methods require no extra time when the rotated solution is already feasible.
>
> This aligns with theory: Eqs. (10)–(11) are closed-form row-wise minimizers, not heuristics, providing stronger per-update decrease and avoiding tuning. We will add this ablation and clarify the rationale.
>
> ---
>
> Q4: Comparison with ANLS-BPP and coordinate descent?
>
> A:  We added ANLS-BPP as a strong baseline (Table 7) , and eNMF remains consistently faster under equal-error: Face (9.8–311.8s vs. 11.4–336.9), Verb (4.93–14.66 vs. 5.58–17.06), Audio (27.8–107.7 vs. 34.8–141.3).
>
> For coordinate descent, we already include A-HALS, a fast HALS/CD solver representative of the same family as scikit-learn’s method. We will clarify this and include the ANLS-BPP results.
>
>
> ---
>
> Q5: Runtime breakdown?
>
> A:  We now provide a phase-wise runtime analysis: (i) SVD initialization, (ii) ADMM rotation, and (iii) feasibility + HALS refinement.
>
> The results show a consistent pattern. On synthetic data (Table 1), eNMF reaches an optimal solution after the first two phases, so runtime is dominated by SVD-computation and rotation. On real datasets (Table 2), the same holds for Audio at higher ranks; otherwise, the final phase remains small relative to total runtime (e.g., Face: 0.88–12.32s vs. 9.8–311.8s; Verb: 0.39–0.73s vs. 4.93–14.66s; Audio low-rank: 1.11–1.63s vs. 27.81–54.43s).
>
> Overall, the runtime advantage comes from the exterior initialization, while the final phase is a lightweight refinement rather than a bottleneck.
>
> ---
>
> W4: F-eNMF vs eNMF unclear
>
> A:  Figure 5 compares two outputs of the same pipeline, both using Algorithm 1. F-eNMF is the feasible solution after the feasibility stage (before refinement), while eNMF is the final result after HALS. The comparison isolates the benefit of the final descent: after feasibility, only a small improvement remains (≤12% error reduction). The caption will be revised accordingly.
>
> W6: missing data
>
> A: We added the requested datasets on the repo.

---

> > ### Author Rebuttal · Reviewer_trzz · 2026-04-04
> >
> > For your reply to Question 3.1: What do you exactly mean by "Grad-Mult"? Please provide the update formulas and parameter choices for this algorithm. Furthermore, Table 5 seems to only provide run-time data. Can you provide experiments that also elucidate the solution quality for some of the test problems? A proper ablation of this should also contain a combination of the proposed feasibility followed by true gradient descent, which is currently lacking.

---

> > > ### Author Response · Authors · 2026-04-07
> > >
> > > Tables are at https://anonymous.4open.science/r/ICML-8F26/README.md.
> > >
> > > Q1: What do you exactly mean by "Grad-Mult"? Please provide the updates and parameter choices for this algorithm.
> > >
> > >
> > > A: By “Grad-Mult”, we refer to a gradient descent method with element-wise adaptive step sizes, which results in multiplicative-form updates that naturally preserve nonnegativity. This approach was originally proposed in [1].
> > > The update rule is given by
> > >
> > > $$V_{bj}^{k+1} = V_{bj}^k - \eta_{bj} (\\nabla\_V f)\_{bj} $$
> > >
> > > The step size is defined as $\eta_{bj}=\frac{\bar V_{bj}^{k}}{((U^{k})^{\top}U^{k}\bar V^{k})_{bj}+\delta}$.
> > >
> > > The auxiliary variable  $\\bar{V}^k $ is defined element-wise as  $$\\bar{V}\_{bj}^k = \\begin{cases} V\_{bj}^k, & \\nabla\_V f \_{bj} \\ge 0 \\\\\\\\ \\max(V\_{bj}^k, \\sigma), & \\nabla\_V f \_{bj} < 0 \\end{cases}$$. We set $\sigma=0.01$ and $\delta=10^{-20}$ to ensure effective gradient scaling and numerical stability.
> > >
> > >
> > > Reference:
> > >
> > > 1.	Lin, C.-J. On the convergence of multiplicative update algorithms for nonnegative matrix factorization. IEEE Transactions on Neural Networks, 18(6):1589–1596, 2007a.
> > >
> > > ---
> > >
> > > Q2: Furthermore, Table 5 seems to only provide run-time data. Can you provide experiments that also elucidate the solution quality for some of the test problems? A proper ablation of this should also contain a combination of the proposed feasibility followed by true gradient descent, which is currently lacking.
> > >
> > > A: Thank you for the helpful follow-up. For completeness, we have now added the two missing gradient-descent-based post-rotation strategies—(iv) Projection + Gradient Descent and (v) Feasibility + Gradient Descent—and updated Table 5 accordingly.
> > > As described in the paper, we benchmark methods under two complementary protocols: an equal-time setup, where competitors are run for the same wall-clock budget as eNMF and the final reconstruction error is compared, and an equal-error setup, where the reconstruction-error target is set to the final error attained by eNMF and competitors are timed until they reach that target. This evaluation protocol is stated explicitly in Section 5.3.
> > > To address the reviewer’s concern that Table 5 originally contained only runtime data, we now provide both runtime and solution-quality evidence for the same five post-rotation strategies:
> > >
> > > (i) Updated Table 5 (equal-error runtime ablation): starting from the same rotated SVD solution, our proposed Feasibility + HALS variant remains the fastest across real datasets. The newly added Feasibility + Gradient Descent variant is also consistently faster than its projection-based counterpart, which further supports our claim that the proposed feasibility-attainment stage preserves useful structure better than direct projection.
> > >
> > > (ii) Table 12 (equal-time relative reconstruction error): under equal-time protocol, our Feasibility + HALS variant achieves the lowest reconstruction error across all datasets and latent dimensions. The projection-based alternatives incur larger relative error, while Feasibility + Gradient Descent is consistently closer to our method than the projection-based variants. This shows that the advantage of the proposed feasibility-attainment stage is not merely runtime, but also better preservation of the rotated low-rank solution under a fixed computational budget.
> > >
> > > (iii) Tables 10 & 11 (downstream solution quality on AudioMNIST): to further elucidate solution quality, we also evaluate the downstream classification performance of the learned NMF factors. Under the equal-error protocol (Table 10), all five post-rotation strategies yield essentially equivalent digit-recognition accuracy, showing that once the same reconstruction target is reached, the final factor quality is very similar. Under the equal-time protocol (Table 11), however, our Feasibility + HALS strategy yields substantially better downstream accuracy,  consistent with Table 12: under limited computation, our method reaches better factor quality faster. At higher ranks ($r=40,80,100$), all methods become identical because the rotated SVD solution itself reaches the unconstrained global minimum, so no post-rotation phase is needed.
> > >
> > > Taken together, these experiments address the reviewer’s concern in two ways: (i) Table 5 is now a complete ablation including the previously missing Feasibility + true Gradient Descent variant, and (ii) solution quality of the different post-rotation strategies is now elucidated not only through equal-error runtime, but also through equal-time reconstruction error (Table 12) and downstream classification performance of the learned NMF factors (Table 10,11).
> > > In summary, we  greatly appreciate the detailed comments and ***have provided multiple experiments to adequately answer their concerns*** across two rounds: these experiments are summarized in Tables 1,2,5,6,7,10,11,12. The additions will improve the revised version’s appeal and we ***kindly*** request the reviewer to adjust their score accordingly.

---

### Official Review · Reviewer_rYxS · 2026-03-13

**Soundness:** 3
**Presentation:** 3
**Significance:** 2
**Originality:** 2
**Overall Recommendation:** 4
**Confidence:** 2

**Summary:**

This paper proposes eNMF, a novel framework for Nonnegative Matrix Factorization (NMF) that departs from traditional interior methods. Instead of enforcing nonnegativity constraints throughout the optimization process, eNMF initializes from the optimal unconstrained factorization (via SVD), employs an ADMM-based rotation procedure to map factors to an exterior point closest to the nonnegative orthant, and subsequently enforces feasibility before descending to a local minimum.Extensive experiments across synthetic and real-world datasets claim significant improvements in reconstruction error and convergence speed compared to 81 baseline configurations.

**Compliance With Llm Reviewing Policy:**

Affirmed.

**Key Questions For Authors:**

Weaknesses:
1.	How does eNMF perform on extremely sparse, high-dimensional datasets where SVD computation becomes the bottleneck?
2.	In the “4.2 Penalty method for attainingf easibility”,this paper introduce $\delta_u$   and $\delta_v$  as penalty parameters.However, there is a lack of theoretical or experiment analysis on how these parameters affect the convergence rate or the quality of the final solution.

**Strengths And Weaknesses:**

Strengths:
This paper proposes a novel framework for Nonnegative Matrix Factorization (NMF) that departs from traditional interior methods. Instead of enforcing nonnegativity constraints throughout the optimization process, eNMF initializes from the optimal unconstrained factorization (via SVD), employs an ADMM-based rotation procedure to map factors to an exterior point closest to the nonnegative orthant, and subsequently enforces feasibility before descending to a local minimum.

Weaknesses:
1.	How does eNMF perform on extremely sparse, high-dimensional datasets where SVD computation becomes the bottleneck?
2.	In the “4.2 Penalty method for attainingf easibility”,this paper introduce $\delta_u$   and $\delta_v$  as penalty parameters.However, there is a lack of theoretical or experiment analysis on how these parameters affect the convergence rate or the quality of the final solution.

---

> ### Author Rebuttal · Authors · 2026-03-31
>
> We thank the reviewer for recognizing the novelty of our eNMF framework, particularly its departure from traditional interior methods through SVD-based initialization, ADMM-driven rotation, and the exterior-to-feasible optimization strategy. The key questions are addressed below. All tables referred in this response can be found at this link https://anonymous.4open.science/r/ICML-8F26/README.md
>
> Q1: How does eNMF perform on extremely sparse, high-dimensional datasets where SVD computation becomes the bottleneck?
>
> A: Thank you for this important question. We directly tested this regime on the Reuters dataset (804414 x 47236, 99.84% sparsity). Even at this scale, eNMF remains faster than the lightweight interior baselines we tested under the equal-error protocol. For r = 10, 20, 40, 80, 100, eNMF takes 41.72, 57.66, 83.04, 126.19, and 152.81 seconds, compared with 44.22-169.62 for ANLS-BPP, 47.56-181.84 for HALS, and 45.47-172.68 for A-HALS. This indicates that, in practice, the exact truncated SVD cost does not erase the wall-clock advantage of eNMF even on an ultra-large and extremely sparse dataset. We will add this result (Table 9) in the revision.
>
> ---
>
> Q2:  In the “4.2 Penalty method for attaining feasibility”,this paper introduce  $\delta_u$ and $\delta_v$ as penalty parameters.However, there is a lack of theoretical or experiment analysis on how these parameters affect the convergence rate or the quality of the final solution.
>
> A: Thank you for this comment. Empirically, we found eNMF to be not very sensitive to the precise choice of $\delta_u$ and $\delta_v$ : we did not require dataset-specific tuning across datasets, latent dimensions, or SNR settings. These parameters affect only the feasibility-attainment stage, whose runtime is a small fraction of the total eNMF runtime; as shown in our added phase-wise table (Tables 1,2), the “Ascent + HALS” component is small across real datasets and zero in several settings. We also find that our feasibility-plus-HALS stage consistently outperforms direct projection (from the infeasible SVD factors) followed by HALS or Grad-Mult (Table 5). We will revise the paper to clarify the role of  $\delta_u$ and  $\delta_v$, and state that we do not currently claim a formal convergence-rate theory for them, and add the empirical evidence showing that performance is robust across the settings considered.

---

> > ### Author Rebuttal · Reviewer_rYxS · 2026-04-04
> >
> > The authors have solved my problems

---

### Official Review · Reviewer_9Jin · 2026-03-13

**Soundness:** 3
**Presentation:** 3
**Significance:** 2
**Originality:** 3
**Overall Recommendation:** 4
**Confidence:** 4

**Summary:**

This paper proposes an exterior framework for Nonnegative Matrix Factorization (eNMF). The method separates low-rank approximation from nonnegativity enforcement by initializing from the optimal unconstrained factorization via truncated SVD. It introduces an ADMM-based rotation procedure to map these unconstrained factors to an exterior point closest to the nonnegative orthant. Finally, it uses an exterior penalty method (Projected Block Coordinate Descent, PBCD) to attain feasibility and employs HALS to descend to a local minimum.
Strengths And Weaknesses

**Compliance With Llm Reviewing Policy:**

Affirmed.

**Final Justification:**

I would maintain my initial score.

**Key Questions For Authors:**

1. Could the authors clarify whether the ADMM rotation step has any known convergence guarantees, or whether it should be viewed purely as a practical heuristic?

2. While Figure 5 provides empirical observations of the error increase during the feasibility enforcement (ascent) stage, is there a provable theoretical upper bound on this degradation? How does the algorithm ensure that this forceful projection does not severely destroy the optimal low-rank structure obtained from the initial SVD?

3. For ultra-large and extremely sparse datasets where exact truncated SVD is computationally prohibitive, how does eNMF's wall-clock time scale compared to lightweight interior methods?

4. Have the authors considered integrating randomized SVD or approximate SVD initializations, and how would that affect the convergence and equivalence claims?

5. The penalty parameters δu and δv  are described as needing to be "sufficiently large" to dominate the gradients. How sensitive is the algorithm's performance to the specific tuning of these hyperparameters across different data distributions in practice?

**Limitations:**

The paper relies on a heuristic ADMM approach for its core rotation step without rigorous theoretical convergence guarantees. The bounds on the error introduced during the feasibility-enforcing ascent stage are not mathematically specified. Additionally, the reliance on exact truncated SVD for initialization may limit the algorithm's scalability on extremely large, sparse datasets.

**Strengths And Weaknesses:**

Strengths:
The paper proposes an exterior-view formulation for NMF that decouples low-rank approximation from nonnegativity enforcement. This perspective is conceptually interesting and differs from traditional interior-update NMF methods.
The use of an orthogonal rotation to align the unconstrained SVD factors with the nonnegative orthant provides an intuitive geometric interpretation.
The experimental evaluation is extensive, comparing against multiple NMF baselines and initialization strategies across both synthetic and real datasets.

Weaknesses:
Heuristic Rotation Step and Missing Theoretical Guarantees. The ADMM procedure used to solve the nonconvex orthogonal Procrustes problem is explicitly described as a heuristic. The paper lacks theoretical convergence guarantees or specific conditions under which this formulation is guaranteed to find the optimal rotation matrix to the positive orthant.
The algorithm utilizes a penalty method (PBCD) to force the exterior point into the feasible region, which the authors acknowledge is an "ascent stage" yielding higher reconstruction error. There is no theoretical upper bound provided to measure how much the optimal low-rank structure obtained from the initial SVD degrades during this forceful projection.

The proposed framework relies on SVD-based initialization and an ADMM-based rotation step, which may introduce additional computational overhead compared to simpler interior-update NMF algorithms. Additionally, the method currently focuses on standard NMF and its applicability to large-scale or streaming settings remains unclear.

---

> ### Author Rebuttal · Authors · 2026-03-31
>
> We thank the reviewer for recognizing the conceptual novelty of our exterior-view formulation, the intuitive geometric insight of the rotation-based approach, and the thoroughness of our experimental evaluation across diverse settings. The key questions are addressed below. All tables referred in this response can be found at this link https://anonymous.4open.science/r/ICML-8F26/README.md
>
> Q1: Could the authors clarify whether the ADMM rotation step has any known convergence guarantees, or whether it should be viewed purely as a practical heuristic?
>
> A: We do not claim a general convergence guarantee for the ADMM rotation step in Algorithm 1; it should be viewed as a practical heuristic for a nonconvex subproblem. This is consistent with the broader setting: NMF is nonconvex and NP-hard, and the orthogonal rotation problem is also nonconvex due to the orthogonality constraint and nonsmooth objective. Accordingly, we do not rely on global convergence guarantees.
>
> That said, the ADMM rotation step is empirically very stable across all tested settings. The returned matrix is almost orthogonal ($\|R^\top R - I\|_F^2 = 0$ on synthetic data and $10^{-5}$–$10^{-14}$ on real data), and convergence is reached in only 1–5 ADMM steps (synthetic) and 2–7 steps (real). These results (see Tables 3,4) indicate that the heuristic is numerically well-behaved and efficient in practice. We will clarify this and include the empirical evidence.
>
> ---
>
> Q2: Is there a theoretical upper bound on error increase during feasibility enforcement, and how is structure preserved?
>
> A: We do not claim a provable upper bound on the reconstruction-error increase and will state this explicitly. Importantly, this stage is not a hard projection onto the nonnegative orthant. Negative entries are pushed upward through penalty-driven updates (Eqs. 6–7), while nonnegative entries are updated via projected block-coordinate descent (Eqs. 8–11) with optimal row-wise step sizes.
>
> This design ensures that feasibility is attained while continuing to reduce reconstruction error on the feasible part, thereby preserving the low-rank structure of the rotated SVD factors rather than overwriting them. Empirically, starting from the same rotated SVD solution, our feasibility+HALS stage is consistently faster than both Projection+HALS and Projection+Grad-Mult across all nontrivial real datasets (see Table 5). This supports our hypothesis that the proposed ascent updates preserve useful structure significantly better than direct projection. We will clarify this point and include this ablation result into the revised manuscript.
>
> ---
>
> Q3: For ultra-large and extremely sparse datasets, how does eNMF scale?
>
> A: We evaluated this regime on the Reuters dataset (804414 × 47236, 99.84% sparsity). Under the equal-error protocol, eNMF remains consistently faster than lightweight interior methods: for $r = 10, 20, 40, 80, 100$, eNMF takes 41.72, 57.66, 83.04, 126.19, and 152.81 seconds, compared with 44.22–169.62 (ANLS-BPP), 47.56–181.84 (HALS), and 45.47–172.68 (A-HALS).
> These results (see Table 9) show that, even at this scale, the cost of truncated SVD does not eliminate the wall-clock advantage of eNMF. We will add this experiment in the revision.
>
> ---
>
> Q4: Can randomized or approximate SVD be integrated?
>
> A: Yes. eNMF only requires an accurate low-rank initialization and does not fundamentally depend on exact truncated SVD, making randomized SVD a natural extension for scalability. The initial rotation and the following steps of eNMF do not make use of the orthogonal structures of $U$ $V$ factors inherent to the exact SVD solutions. We performed new experiments on all the real data sets (see Table 8) and randomized SVD yields essentially identical end-to-end runtime to exact truncated SVD on these benchmarks, indicating that eNMF is robust to replacing the exact SVD with a high quality approximate low rank initializer.
>
> ---
>
> Q5: Sensitivity to $\delta_u$ and $\delta_v$?
>
> A: Empirically, eNMF is not sensitive to these parameters: we did not require dataset-specific tuning across datasets, latent dimensions, or SNR levels. Across all settings, the ascent + PBCD stage reaches local minima quickly, and the feasibility+HALS procedure consistently outperforms projection-based alternatives (Projection+HALS and Grad-Mult, see Table 5). While we do not provide a formal sensitivity bound, the results indicate strong empirical robustness. We will clarify this in the revision.

---

> > ### Author Rebuttal · Reviewer_9Jin · 2026-04-04
> >
> > The author partially solved my problem, therefore I uphold the original judgment.

---

> > > ### Author Response · Authors · 2026-04-07
> > >
> > > Tables are at https://anonymous.4open.science/r/ICML-8F26/README.md.
> > >
> > > We thank the reviewer again for the thoughtful feedback. In response to the concerns raised in the initial review, we did our best in the first-round rebuttal to address each of the reviewer’s points. Because the follow-up comment indicates that some concerns may still remain, without specifying which ones, we have proactively added several new experiments and ablations to address the likely remaining issues as thoroughly as possible.
> > > In our previous response, we already added: (i) empirical convergence evidence for the ADMM rotation step, showing that it returns an almost perfectly orthogonal matrix and converges in very few iterations across both synthetic and real datasets (Tables 3, 4); and (ii) additional scalability experiments, including Reuters, ANLS-BPP comparisons, and randomized-SVD experiments, to address questions about large-scale sparse settings and approximate low-rank initialization (Tables 8, 9).
> > > In this revision, we further added:
> > >
> > > - a detailed ablation of the feasibility-attainment stage, now including the previously missing gradient-descent-based variants, showing that our proposed feasibility-based updates are consistently faster than projection-based alternatives when starting from the same rotated SVD point (Table 5);
> > >
> > > - solution-quality comparisons in addition to runtime, including equal-time relative reconstruction error and downstream classification performance of the learned NMF factors, which show that our method yields better solutions under equal-time budgets and comparable solutions under equal-error budgets (Tables 10, 11, 12).
> > >
> > >
> > > Taken together, these additional results support the same conclusion as in the original submission: the main advantage of eNMF comes from the exterior rotation plus feasibility-attainment strategy, which is not only computationally efficient but also preserves solution quality better than simpler projection-based alternatives.
> > > If any concern still remains insufficiently addressed, we would be very grateful for the opportunity to clarify it further.

---

### Decision · Program_Chairs · 2026-04-30

**Decision:**

Accept (regular)

**Comment:**

This paper proposes the eNMF algorithm, a new method for efficiently computing NMF. It consists of (1) an initialization stage with SVD and rotation, and (2) a refinement stage consisting of penalty-based optimization. This natural initialization + refinement algorithm allows infeasible (i.e., exterior) iterates, unlike the popular feasible (i.e., interior) methods. Its primary strengths lie in its superior convergence speed and solution quality, achieving up to 150% speedup and a 30% reduction in reconstruction error while remaining robust across diverse datasets and initialization schemes. Reviewers praised its intuitive geometric interpretation and extensive empirical validation, including scalability on ultra-large sparse datasets. However, weaknesses include a lack of theoretical convergence guarantees for the heuristic ADMM rotation step and the absence of a mathematical upper bound on error degradation during feasibility enforcement. While authors addressed concerns regarding hyperparameter sensitivity and runtime breakdowns through additional ablations, the method’s reliance on SVD-based initialization remains a potential bottleneck for specific large-scale applications.